# STT-LLM: Structural-Temporal Tokenization for Adapting LLMs to Longitudinal Clinical Profiles

**Maxx Richard Rahman** [1 2]  **Mostafa Hammouda** [1 2]  **Wolfgang Maass** [1 2]

## Abstract

Large Language Models have shown strong generalization across natural language tasks but remain underexplored for longitudinal clinical profiles. In sports anti-doping, biological profiles are analyzed to support early detection of prohibited substance use and identification of anomalous biological patterns, both of which require joint modeling of temporal dynamics and metabolic relationships. We propose STT-LLM, a structural-temporal tokenization framework that adapts LLMs to longitudinal clinical analysis without modifying their backbone architectures. STT-LLM constructs biologically grounded structural-temporal embeddings and transforms them into LLM-compatible tokens via specialized tokenizers that explicitly encode pathway structure and temporal evolution. We evaluate STT-LLM on real-world longitudinal datasets from athletes, showing consistent improvements over native LLM tokenization strategies in sequence prediction and anomaly detection. In addition, we present a case study where STT-LLM provides contextual reasoning that aligns more closely with expert assessments compared to baseline models. These results highlight tokenization as a key bottleneck and opportunity for adapting LLMs to clinical data.

## 1. Introduction

Large Language Models (LLMs) have shown strong generalization across natural language and multimodal tasks, including reasoning and code generation (Chang et al., 2024; Matarazzo & Torlone, 2025). This success has motivated growing interest in adapting LLMs to scientific and biomedical domains, where data are often structured, scarce, and heterogeneous. A particularly challenging setting arises in longitudinal clinical analysis, such as sports doping control, where athletes' steroid profiles are monitored over time to detect prohibited substance use. These longitudinal profiles represent individual-specific trajectories that are irregularly sampled and governed by underlying metabolic pathways (Schüssler-Fiorenza Rose et al., 2019). Accurate modeling of such data requires sensitivity to both temporal dynamics and biochemical structure, which are important for distinguishing natural physiological variation from doping-induced abnormalities.

However, most LLMs are pretrained on unstructured text corpora and operate on discrete token sequences, which are poorly aligned with time-dependent numerical biomedical signals (Raiaan et al., 2024; Naveed et al., 2023). Naïve adaptations, such as flattening profiles into text or tabular prompts, discard structural relationships among metabolites and fail to encode pathway-level constraints that govern longitudinal evolution (Das et al., 2025). While recent works show that LLMs can process clinical time-series through prompting or embedding reprogramming (Chan et al., 2024; Xiao et al., 2025), these approaches largely treat temporal signals as flat sequences and lack inductive biases for multivariate irregular sampling. In parallel, domain-specific models such as graph neural networks and time-series transformers explicitly encode structure or temporal dependencies (Ghanvatkar & Rajan, 2023; Luo et al., 2024a; Tipirneni & Reddy, 2022; Xu et al., 2023), but their specialized architectures are not directly compatible with general-purpose LLM inference pipelines, limiting their flexibility for multitask or instruction-driven settings. These limitations are further exacerbated in anti-doping scenarios, where athlete profiles are extremely sparse and heterogeneous, often consisting of only one or two samples collected at irregular intervals (Lauritzen & Solheim, 2024).

In this paper, we argue that tokenization (not model architecture) is the central bottleneck in adapting LLMs to longitudinal biomedical analysis. We introduce STT-LLM, a structural-temporal tokenization framework that directly incorporates metabolic pathway structure and irregular temporal dynamics into LLM-compatible tokens. STT-LLM

---

[1]German Research Center for Artificial Intelligence (DFKI), Saarbrücken, Germany [2]Saarland University, Saarbrücken, Germany. Correspondence to: Maxx Richard Rahman <maxx_richard.rahman@dfki.de>.

*Proceedings of the 43rd International Conference on Machine Learning*, Seoul, South Korea. PMLR 306, 2026. Copyright 2026 by the author(s).

constructs joint structural-temporal embeddings from longitudinal profiles and transforms them via specialized tokenizers, enabling pretrained LLMs to operate on biologically grounded representations without modifying their backbone architectures. This design allows LLMs to support sequence prediction for early detection of prohibited substance use and anomaly detection under extreme data scarcity, while retaining the reasoning capability and extensibility of general-purpose LLMs. The key contributions of this work are:

- We introduce a structural-temporal tokenization framework that adapts general-purpose LLMs to longitudinal clinical profiles by explicitly encoding metabolic pathway structure and irregular temporal dynamics.

- We propose specialized structural and temporal tokenizers that transform biologically grounded embeddings into LLM-compatible tokens, enabling domain adaptation without modifying the LLM backbone.

- We show the effectiveness of our work on real-world steroid profiles, where it consistently outperforms native LLM tokenization strategies in both sequence prediction and anomaly detection.

## 2. Related Works

**Tokenization and Embedding for Domain Adaptation.** Tokenization defines how input signals are mapped to the representational space of LLMs, yet most existing schemes are designed for natural language and do not generalize to structured longitudinal data. Classical tokenizers such as byte-pair encoding and WordPiece (Sennrich et al., 2015; Wu et al., 2016) operate on discrete text units and cannot encode temporal irregularity, or relational structure. Prior work has explored task-aware tokenization and embedding strategies for domain adaptation (Huang et al., 2025; Liu et al., 2024a), including tabular formats (TAPEX (Liu et al., 2021a), TabLLM (Hegselmann et al., 2023)) and graph-aware prompting mechanisms (GraphPrompt (Sun et al., 2022), Graph-of-Thought (Besta et al., 2024)). While effective for static or relational inputs, these methods do not jointly model temporal evolution and domain-specific structure. Other embedding approaches, such as distance-based transfer (Liu et al., 2021b) or hypernetwork-generated tokens (Feher et al., 2024), rely on auxiliary models and task-specific heuristics, limiting their applicability in settings where biological constraints and temporal progression must be encoded directly. In contrast, doping monitoring requires representations that simultaneously preserve metabolic pathway structure and longitudinal dynamics.

**LLMs for Longitudinal Modeling.** Several recent studies have investigated adapting pretrained LLMs to time-series analysis via prompt augmentation (Rahman et al.,

2024). Methods such as Time-LLM (Jin et al., 2023) and UniTime (Liu et al., 2024b) project temporal patches into token sequences, demonstrating that LLMs can model time-indexed data without architectural changes. However, these approaches typically flatten multivariate signals and treat time as an auxiliary index, which limits their ability to capture fine-grained temporal irregularities and cross-variable dependencies. In biomedical domains, related efforts include timeline extraction (Frattallone-Llado et al., 2024), event ordering (Leeuwenberg & Moens, 2020), and hybrid models combining clinical text with structured measurements (Jeong et al., 2024; Belyaeva et al., 2023), but these methods often depend on fixed annotations or handcrafted features (Hou et al., 2020; Noroozizadeh et al., 2023). As a result, general-purpose LLMs remain poorly aligned with highly sparse, individual-specific longitudinal profiles, such as those encountered in anti-doping, where rare anomalies must be detected under zero- or few-shot conditions.

## 3. Problem Formulation

We consider a longitudinal clinical profile consisting of repeated multivariate measurements over time. For athlete $i$, the profile is represented as $\mathbf{X}_i = [\mathbf{x}_{ij}]_{j=1}^{n_i} \in \mathbb{R}^{p \times n_i}$ ($p$: number of clinical parameters, $n_i$: number of samples). Each observation $\mathbf{x}_{ij} \in \mathbb{R}^p$ corresponds to the $j^{th}$ time point. Structural dependencies among parameters are encoded by a feature interaction graph $A \in \mathbb{R}^{p \times p}$ ($A_{k,l}$: interaction between parameters $k$ and $l$).

**Sequence Prediction** Given observations up to time $t$, denoted by $\mathbf{X}_{i,1:t}$, the goal is to predict the next observation $\hat{\mathbf{x}}_{i,t+1} = f_\theta(\mathbf{X}_{i,1:t}, A)$, where $f_\theta$ models both temporal dependencies across samples and structural dependencies induced by $A$.

**Anomaly Detection** Anomaly detection is performed at both the sample and profile levels. A *local* anomaly score for sample $j$ is defined as $s_{ij}^{\text{local}} = g_\phi(\mathbf{x}_{ij}, \hat{\mathbf{x}}_{ij})$ ($\hat{\mathbf{x}}_{ij}$: model-predicted observation, $g_\phi$: scoring function). Sample $j$ is flagged as anomalous if $s_{ij}^{\text{local}} > \epsilon_{\text{local}}$. The *global* anomaly score for profile $i$ is defined as $s_i^{\text{global}} = s_{i,n_i}^{\text{local}}$, corresponding to the most recent observation. The profile is classified as anomalous if $s_i^{\text{global}} > \epsilon_{\text{global}}$.

## 4. STT-LLM: Structural-Temporal Tokenization for Large Language Models

### 4.1. Input Prompt

As shown in Fig. 1, the input prompt $\mathbf{I}$ consists of two components: the task $\mathbf{P}$, which is a textual description providing instructions, and the longitudinal profile $\mathbf{X}_i$. The task prompt $\mathbf{P}$ is processed using a pre-trained language

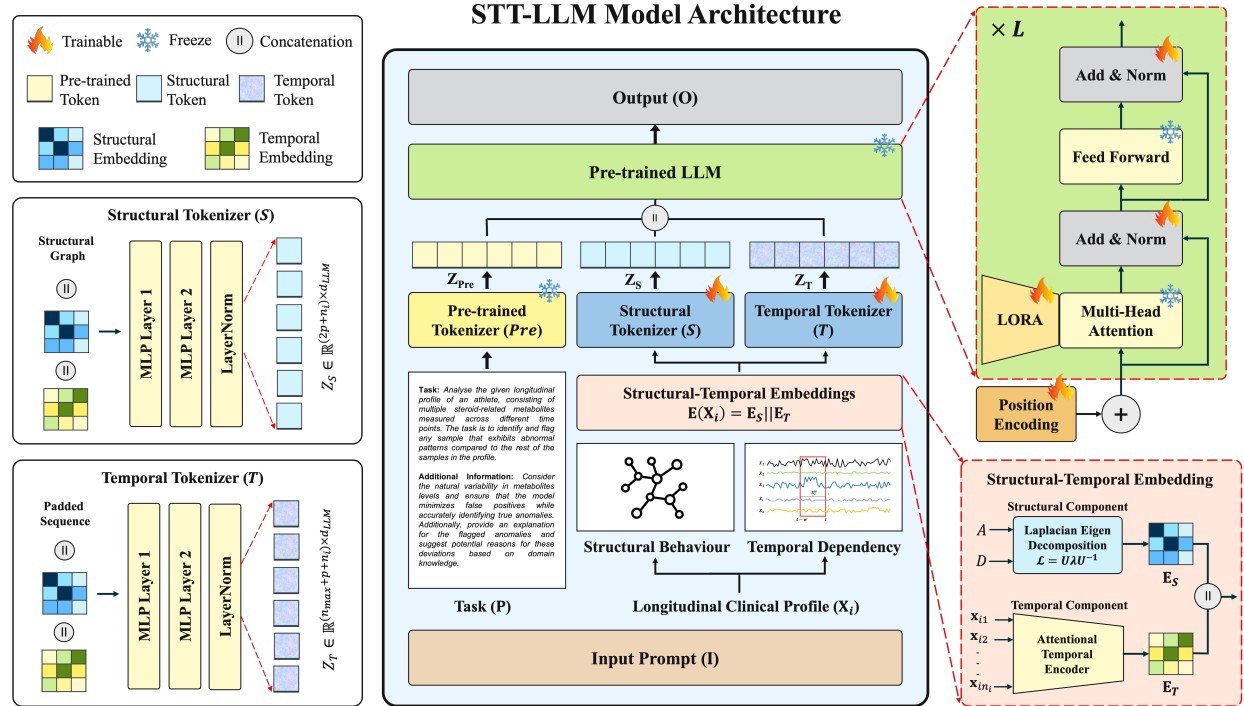

*Figure 1.* Proposed model architecture of STT-LLM for analyzing longitudinal clinical profile.

tokenizer to produce token embeddings $Z_{\text{Pre}}$, while $\mathbf{X}_i$ is fed into the proposed tokenization framework that integrates structural and temporal dependencies. This dual processing strategy enables the model to align semantic task instructions with rich domain-specific data representations.

## 4.2. Structural-Temporal Embeddings

**Structural Component.** Given an adjacency matrix $A$ and a degree matrix $D$ of feature interaction graph, the normalized graph Laplacian $\mathcal{L} = I - D^{-\frac{1}{2}}AD^{-\frac{1}{2}}$ ($I$: identity matrix, $D$: node degrees) (Kipf & Welling, 2017). This normalized Laplacian $\mathcal{L}$ encodes important structural properties (connectivity, community). The eigen-decomposition can be calculated as $\mathcal{L} = U\lambda U^{-1}$ ($U$: eigenvectors, $\lambda$: eigenvalues). To obtain the structural embedding, the eigenvectors are projected through a learnable transformation: $\mathbf{E}_S = W_{\mathbf{E}_S}U + b_{\mathbf{E}_S}$ ($W_{\mathbf{E}_S}$, $b_{\mathbf{E}_S}$: trainable parameters). When domain priors are unavailable, $A$ can be replaced with a correlation-based graph or identity matrix.

**Temporal Component.** The temporal behavior is modeled using an attention mechanism as $\text{Attention}(Q, K, V) = \text{softmax}\left(\frac{QK^T}{\sqrt{d_k}}\right)V$, where $Q$, $K$, $V$ are linear projections (Vaswani et al., 2017). To incorporate temporal order, positional encodings are added, defined as $PE_{(pos,2i)} = \sin\left(\frac{pos}{10000^{2i/d}}\right)$ and $PE_{(pos,2i+1)} = \cos\left(\frac{pos}{10000^{2i/d}}\right)$ ($pos$: position, $i$: dimen-

sion index). These encodings allow the model to distinguish between positions in the input sequence. The attention output $\mathbf{A}_T$, is passed through a feed-forward network to produce $Z_{ST} = \text{ReLU}(\mathbf{A}_T W_{\mathbf{E}_{T_1}} + b_{\mathbf{E}_{T_1}})W_{\mathbf{E}_{T_2}} + b_{\mathbf{E}_{T_2}}$ and layer normalization is applied to produce $\mathbf{E}_T = \text{LayerNorm}(Z_{ST})$. This architecture stabilizes training and facilitates gradient flow. The resulting temporal embeddings $\mathbf{E}_T$ capture dynamic patterns important for modeling longitudinal profiles. Finally, the structural and temporal embeddings are concatenated to form the unified structural-temporal embedding $\mathbf{E}(\mathbf{X}_i) = \mathbf{E}_S \| \mathbf{E}_T \in \mathbb{R}^{(p+n_i)\times p}$. This joint embedding ensures comprehensive integration of structural and temporal information, preparing the longitudinal clinical data for tokenization.

### 4.3. Tokenization

**Structural Tokenizer** ($S$). The framework processes the structural aspects of the structural-temporal embeddings by effectively encoding a feature interaction graph constructed from domain knowledge in a longitudinal profile. The input structural representation $A$ of longitudinal profile is combined with the learned structural-temporal embedding $\mathbf{E}(\mathbf{X}_i)$, yielding the concatenated input $\mathbf{X}_S = A \| \mathbf{E}(\mathbf{X}_i)$, $\mathbf{X}_S \in \mathbb{R}^{(2p+n_i)\times p}$. The concatenated input $\mathbf{X}_S$ is then processed through a multi-layer perceptron (MLP) with two layers. The first layer applies a

ReLU nonlinearity $H_S = \text{ReLU}(\mathbf{X}_S W_{S_1} + b_{S_1})$, $H_S \in \mathbb{R}^{(2p+n_i) \times d_{\text{hidden}}}$, followed by a linear transformation $Z_S^{MLP} = H_S W_{S_2} + b_{S_2}$, $Z_S^{MLP} \in \mathbb{R}^{(2p+n_i) \times d_{LLM}}$, where $W_{S_1} \in \mathbb{R}^{p \times d_{\text{hidden}}}$, $W_{S_2} \in \mathbb{R}^{d_{\text{hidden}} \times d_{LLM}}$ ($b_{S_1}$, $b_{S_2}$: trainable parameters). To ensure stable training and consistent scaling of the token embeddings, layer normalization is applied $Z_S = \text{LayerNorm}(Z_S^{MLP})$, $Z_S \in \mathbb{R}^{(2p+n_i) \times d_{LLM}}$, where $d_{LLM}$ is the target embedding dimension compatible with the downstream LLM. The resulting structural token embeddings $Z_S$ encode both the structural relationships captured by the graph and the dynamic patterns captured by the structural-temporal embeddings.

**Temporal Tokenizer** ($T$). To handle sequences heterogeneity, we apply i) *Padding:* Sequences shorter than $n_{\max}$ are zero-padded to ensure uniform input dimensions and ii) *Masking:* Mask $M \in \mathbb{R}^{n_{\max}}$ indicates valid time steps, with marked real samples: 1 and padded samples: 0. This ensures the model focuses computations on valid temporal entries. The padded temporal sequence $\mathbf{X}_T^{\text{padded}}$ is combined with the structural-temporal embedding $\mathbf{E}(\mathbf{X}_i)$ $\mathbf{X}_T = \mathbf{X}_T^{\text{padded}} || \mathbf{E}(\mathbf{X}_i)$, $\mathbf{X}_T \in \mathbb{R}^{(n_{\max}+p+n_i) \times p}$, where $n_{\max}$ is the maximum sequence length of longitudinal profile in the dataset. The concatenated temporal input is passed through a two-layer MLP. The first layer applies a nonlinear transformation $H_T = \text{ReLU}(X_T W_{T_1} + b_{T_1})$, $H_T \in \mathbb{R}^{(n_{\max}+p+n_i) \times d_{\text{hidden}}}$, followed by a second linear layer $Z_T^{\text{MLP}} = H_T W_{T_2} + b_{T_2}$, $Z_T^{\text{MLP}} \in \mathbb{R}^{(n_{\max}+p+n_i) \times d_{LLM}}$. While the MLP processes the entire longitudinal profile, the mask $M$ ensures only valid time steps influence the learned embeddings. Finally, layer normalization is applied to stabilize learning and ensure consistent scaling $Z_T = \text{LayerNorm}(Z_T^{\text{MLP}})$, $Z_T \in \mathbb{R}^{(n_{\max}+p+n_i) \times d_{LLM}}$. The resulting temporal token embeddings $Z_T$ capture the temporal evolution while preserving structural context.

### 4.4. Model Training

The output token embeddings $Z_{\text{Pre}}$, $Z_S$, and $Z_T$ are concatenated $\mathbf{Z} = Z_{\text{Pre}} || Z_S || Z_T$, where $\mathbf{Z} \in \mathbb{R}^{L \times d_{LLM}}$, with $L$ denoting the token sequence length and $d_{LLM}$ the embedding dimension compatible with the LLM backbone. This combined representation is passed to a pre-trained LLM, which has been augmented with LoRA adapter $\mathbf{O} = \text{Adapter}(\text{LLM})$, where $\mathbf{O}$ represents the model output for different downstream tasks, such as sequence prediction and anomaly detection over longitudinal profiles. During training, the tokenizers ($S$, $T$) are trained jointly with the LoRA adapter, while the core LLM weights remain frozen. This setup allows efficient adaptation to specific downstream tasks with minimal computational overhead, leveraging the generalization capabilities of the pre-trained LLM while enabling domain-specific adaptation through the tokenizers and LoRA layers. The training objective functions can be

*Table 1.* Summary statistics of all the datasets.

| Datasets | Gender | # Profiles | # Samples | Length $n_i$ |
|---|---|---|---|---|
| Steroid-M | Male | 755 | 4214 | 3-20 |
| Steroid-F | Female | 375 | 2307 | 3-20 |
| Steroid-M$_{\text{lim}}$ | Male | 737 | 1474 | 2 |
| Steroid-F$_{\text{lim}}$ | Female | 293 | 586 | 2 |

defined according to the downstream task.

## 5. Experiments

**Datasets.** All the models are evaluated on real-world athlete datasets (Table 1) consisting of longitudinal steroid profiles derived from the urine samples (Rahman et al., 2022). The dataset includes measurements of six key steroid metabolites: testosterone (T), epitestosterone (E), etiocholanolone (Etio), androsterone (A), $5\alpha$-androstanediol ($5\alpha$Adiol), and $5\beta$-androstanediol ($5\beta$Adiol) following the steroid metabolism pathway to synthesize (Piper et al., 2021). The profile lengths range from 2-20 samples per athlete, reflecting realistic variability in longitudinal monitoring. These datasets cover diverse population groups and temporal resolutions, allowing us to comprehensively evaluate STT-LLM under realistic conditions.

**Baselines.** We compare the STT-LLM tokenization strategy against different mid-sized LLMs, including Qwen-2.5 (7B) (Yang et al., 2025), Falcon-3 (7B) (Almazrouei et al., 2023), Mistral (7B) (Jiang et al., 2023), LLaMA-2 (7B) (Touvron et al., 2023), LLaMA-3.1 (8B) (Grattafiori et al., 2024), Phi-4 (7B) (Abdin et al., 2024), and DeepSeek-R1 (7B) (DeepSeek-AI et al., 2025). Each model is fine-tuned on different downstream tasks using its native tokenization strategy.

**Baseline Serialization.** For all baseline LLMs using native tokenization, longitudinal profiles are serialized into text sequences by concatenating all time steps per subject into a single structured text prompt (via row ordering), which is then tokenized using each model's pretrained tokenizer without modification. While this approach is simple and generalizable, it suffers from two key limitations: (i) byte-pair encoding fragments numerical values unpredictably, and (ii) no explicit temporal order or structural relationship among variables is encoded. Empirically, all seven baselines using this approach show degenerate anomaly detection behavior. STT-LLM similarly generalizes to any multivariate time-series with inter-variable dependencies, with structural gains proportional to the availability of domain priors for constructing the feature interaction graph.

**Experimental Setup.** The evaluation was performed under two settings: *zero-shot*, and *few-shot* (2-20 labeled exam-

ples as in-context prompts). The evaluation metrics used are RMSE, MAE, and MAPE for sequence prediction, and accuracy, sensitivity, precision, F1-score, and AUC for anomaly detection. We set the high specificity value (99.9%) to avoid any false positives (domain requirements). All reported results are averaged over three independent runs with standard deviations reported where applicable.

STT-LLM uses LLaMA-3.1 (8B) as its backbone LLM in all main experiments unless otherwise stated. To verify that observed gains are not specific to this choice, Figure 4 compares native tokenization against STT-LLM tokenization across all seven backbone LLMs evaluated in this study. As shown, STT-LLM consistently improves both anomaly detection sensitivity and sequence prediction RMSE across every backbone, with sensitivity gains ranging from $+2$ to $+14$ percentage points and RMSE reductions ranging from 9.35% to 26.26%. This confirms that performance improvements stem from the tokenization framework rather than the choice of backbone architecture.

# 6. Results

## 6.1. Sequence Prediction

**Zero-shot setting.** Fig. 2 shows that STT-LLM consistently outperforms all LLM baselines by achieving the lowest error scores. For Steroid-M and Steroid-F, STT-LLM reduces RMSE value (%100) to 79.3 and 68.4, respectively, while all baselines remain above 83, indicating its improved ability to model metabolic patterns even without supervision. The gains are even more pronounced in the limited datasets, where STT-LLM achieves low RMSE value (%100) of 30.0 and 1.2, respectively, outperforming the next-best models by large margins. For MAE value (%10), STT-LLM consistently achieves the lowest errors across datasets, with values dropping to near 5-6 on the limited datasets, reflecting accurate point-wise predictions.

**Few-shot setting.** Table 8 reports few-shot sequence prediction performance across all models. Increasing the number of in-context examples from 5 to 20 does not consistently reduce error and often leads to higher RMSE and MAE across all baselines. Across all datasets and shot counts, STT-LLM achieves the lowest RMSE, indicating more robust temporal generalization. In particular, at 5-shot, STT-LLM attains the lowest RMSE on Steroid-$M_{lim}$ (1730.11) and Steroid-$F_{lim}$ (1276.32), and similarly yields the lowest MAE on Steroid-$F_{lim}$ at 10-shot (643.71) and 20-shot (642.90). Error metrics for STT-LLM remain stable across shot settings, whereas baselines exhibit higher fluctuations, showing improved robustness to prompt variability.

Qwen-2.5 produces identical predictions across all shot settings on all four datasets (e.g., RMSE $=$ 1695.99 regardless

of shot count on Steroid-M). This is not a reporting error but rather reflects a model-level failure: Qwen-2.5 was unable to condition its outputs on the structured numerical in-context examples provided, producing effectively constant predictions irrespective of the few-shot context. This behavior is itself consistent with our hypothesis, i.e., native text tokenization fails to encode structured longitudinal data in a way that enables LLMs to leverage few-shot supervision effectively.

## 6.2. Anomaly Detection

**Zero-shot setting.** Table 3 reports zero-shot anomaly detection performance for both local and global tasks. For local anomaly detection, STT-LLM achieves substantially higher sensitivity than all baselines, reaching 15.0% on Steroid-M and 17.0% on Steroid-$F_{lim}$, whereas most LLM baselines exhibit near-zero sensitivity despite high accuracy. This behavior indicates a degenerate decision regime in which baselines predominantly predict the majority (normal) class. In contrast, STT-LLM maintains non-trivial sensitivity with moderate accuracy (87-88%), yielding improved precision and F1-scores. For global anomaly detection, performance improves across all models due to reduced sparsity. STT-LLM attains the highest F1-scores (0.26 on Steroid-M, 0.29 on Steroid-F) and AUC values (0.57 and 0.59, respectively), exceeding baseline models by up to $\sim$10%. These results show that STT-LLM better separates anomalous and normal profiles under zero-shot conditions, particularly in rare-event regimes where sensitivity and ranking performance are critical.

**Few-shot setting.** Fig. 3 shows that STT-LLM exhibits consistent performance gains in global anomaly detection as the number of in-context examples increases. On Steroid-M, sensitivity improves monotonically from 0.15 (2-shot) to 0.60 (20-shot), while precision increases across all datasets, reaching near-saturation on Steroid-F and Steroid-$F_{lim}$. Correspondingly, F1-scores increase substantially (e.g., 0.15 to 0.70 on Steroid-M), indicating balanced improvements in both recall and precision. In contrast, baseline LLMs display non-monotonic or unstable trends across shot settings, suggesting limited ability to effectively leverage few-shot supervision. Paired $t$-tests across three runs confirm that F1 improvements over the best baseline reach statistical significance ($p < 0.05$) at 5-shot and above. At 2-shot, improvements are directionally consistent ($p = 0.08$).

Table 4 reports the performance of representative non-LLM baselines, including classical time-series and graph-based models, which are often strong when tailored to a single task. However, these models rely on specialized architectures and task-specific training pipelines, making them difficult to integrate into a unified or multi-task framework. In contrast, STT-LLM achieves competitive performance while operat-

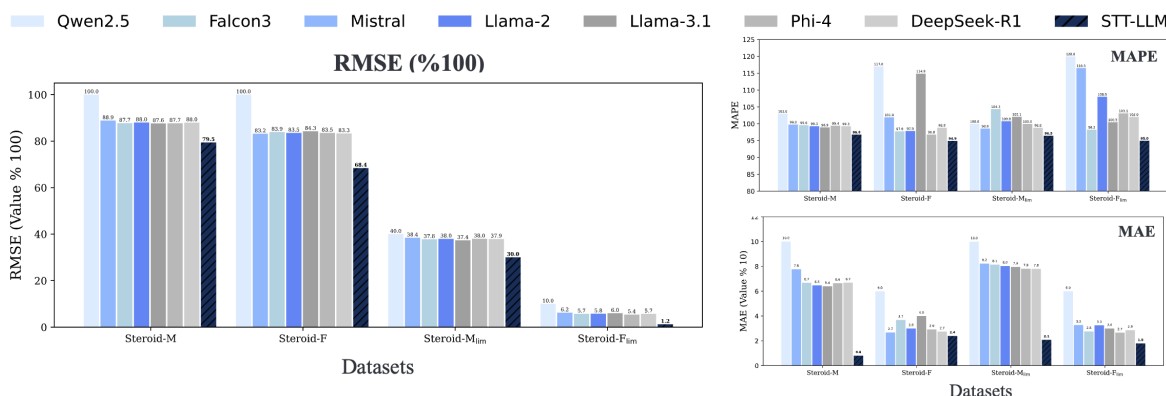

*Figure 2.* Zero-shot sequence prediction performance across different datasets.

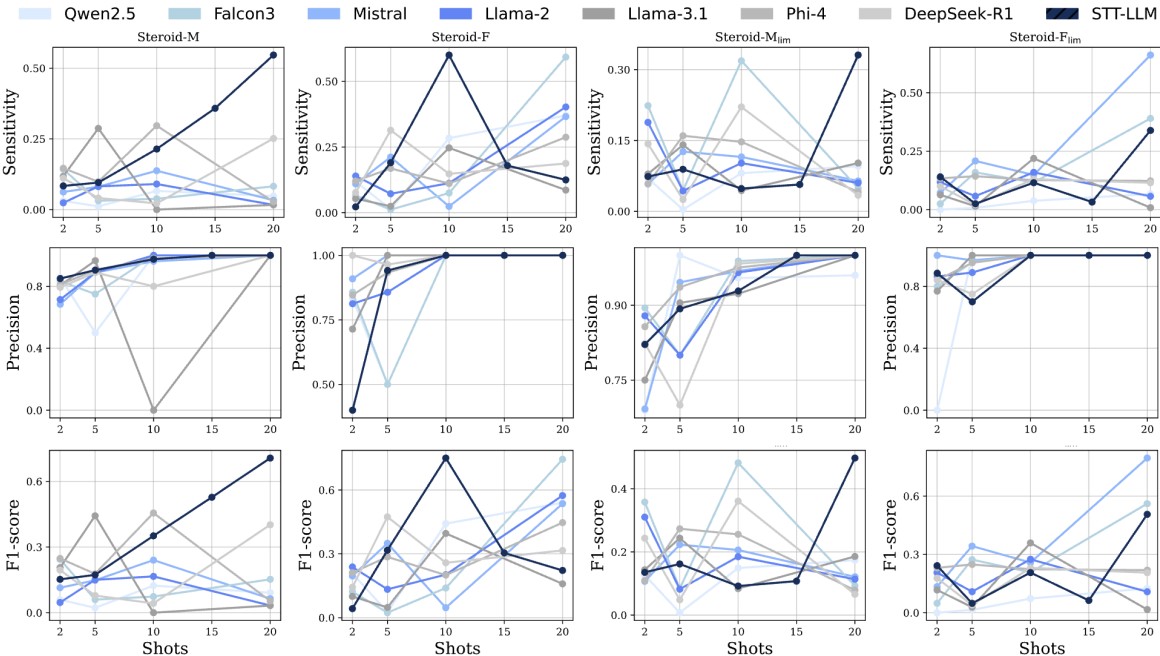

*Figure 3.* Few-shot global anomaly detection performance across different metrics.

ing entirely within an LLM-compatible inference pipeline, underscoring that our goal is not to outperform all specialized models in isolation, but to enable flexible longitudinal modeling using general-purpose LLMs.

### 6.3. Ablation Study

The ablations include removing all components (*w/o all*), structural tokenizer (*w/o structural*), temporal tokenizer (*w/o temporal*), embedding layer (*w/o embeddings*), and pairs of components. Table 5 shows that STT-LLM achieves the lowest sequence prediction errors (RMSE: 1664.59, MAPE: 96.80). Removing all components increases RMSE: +1.4%, MAE: +1.7%, MAPE: +2.2% relative to STT-LLM. Removing embeddings alone increases MAPE to 100.56 (+3.9%) and drops AUC to 0.5352 (-5.7%), highlighting the

embedding layer's key role in aligning multimodal representations. For anomaly detection, STT-LLM achieves a good balance across different metrics. Removing all components lowers sensitivity by -27.8%, and precision by -20.5% compared to STT-LLM. Removing either the structural or temporal tokenizer reduces sensitivity by -50% (0.0968 - 0.1237) and precision by -33% (0.2769 - 0.2987), showing that both structural and temporal components are important for anomaly detection. When two components are removed, the degradation is even sharper, e.g., *w/o embeddings + structural* drops AUC by -14% (0.4887) relative to STT-LLM. The *w/o all* variant performs slightly better than some partial ablations because complete removal avoids embedding mismatches and produces uniform flat inputs, whereas partial removal yields incoherent fused representations that

*Table 2.* Few-shot sequence prediction results across different datasets.

| Datasets | Baseline | @5 | | | @10 | | | @15 | | | @20 | | |
|---|---|---|---|---|---|---|---|---|---|---|---|---|---|
| | | RMSE↓ | MAE↓ | MAPE↓ | RMSE↓ | MAE↓ | MAPE↓ | RMSE↓ | MAE↓ | MAPE↓ | RMSE↓ | MAE↓ | MAPE↓ |
| Steroid-M | Qwen-2.5 | 1695.99 | 899.99 | 111.99 | 1695.99 | 899.99 | 111.99 | 1695.99 | 899.99 | 111.99 | 1695.99 | 899.99 | 111.99 |
| | Mistral | 1688.34 | 894.92 | 98.19 | 1690.63 | 896.48 | **94.69** | 1688.90 | 896.54 | 101.04 | 1692.39 | 899.84 | 110.19 |
| | Falcon-3 | 1688.02 | 896.54 | 101.17 | 1689.88 | 897.20 | 100.31 | 1690.39 | 897.48 | 100.01 | 1691.48 | 897.69 | 100.93 |
| | LLaMA-2 | 1687.80 | 895.81 | 98.21 | 1689.47 | 897.29 | 100.74 | 1690.01 | 896.86 | 100.47 | 1691.16 | 897.06 | 98.19 |
| | LLaMA-3.1 | 1688.57 | 896.78 | 100.27 | 1689.67 | 898.19 | 106.75 | 1690.56 | 897.17 | 101.46 | 1690.98 | 896.84 | 97.21 |
| | Phi-4 | 1688.20 | 896.62 | 100.38 | 1690.17 | 897.21 | 97.98 | 1690.04 | 897.36 | 102.87 | 1691.41 | 897.54 | 100.88 |
| | DeepSeek-R1 | 1688.05 | 896.73 | 100.33 | 1689.88 | 896.73 | 98.31 | 1690.31 | 897.17 | 98.81 | 1691.65 | 897.56 | 99.48 |
| | **STT-LLM** | **1680.00** | **890.77** | **96.80** | **1681.57** | **891.27** | 96.79 | **1682.06** | **891.37** | 96.79 | **1683.51** | **891.87** | 96.81 |
| Steroid-F | Qwen-2.5 | 1395.99 | 699.99 | 129.99 | 1395.99 | 699.99 | 129.99 | 1395.99 | 699.99 | 129.99 | 1395.99 | 699.99 | 129.99 |
| | Mistral | 1387.98 | 695.23 | 120.35 | 1392.05 | 697.48 | **92.08** | 1388.75 | 695.68 | 108.68 | 1390.98 | 696.63 | 107.37 |
| | Falcon-3 | 1388.12 | 695.07 | 93.71 | 1389.62 | 695.41 | 93.80 | 1388.77 | 695.67 | 115.24 | 1389.53 | 694.61 | **94.31** |
| | LLaMA-2 | 1387.93 | 694.93 | **93.30** | 1388.86 | 695.52 | 109.01 | 1388.92 | 694.91 | 100.61 | 1389.93 | 695.33 | 101.76 |
| | LLaMA-3.1 | 1388.67 | 695.34 | 94.12 | 1389.38 | 695.03 | 93.93 | 1388.72 | 695.19 | 106.55 | 1390.03 | 695.23 | 103.50 |
| | Phi-4 | 1388.07 | 695.39 | 98.83 | 1389.09 | 695.64 | 108.44 | 1389.50 | 694.70 | 99.72 | 1389.75 | 695.43 | 108.37 |
| | DeepSeek-R1 | 1388.48 | 695.47 | 102.55 | 1389.54 | 696.04 | 97.25 | 1389.05 | 695.14 | 98.93 | 1389.09 | 694.41 | 95.36 |
| | **STT-LLM** | **1372.85** | **684.39** | 94.94 | **1374.17** | **684.89** | 94.92 | **1373.51** | **684.05** | 94.91 | **1374.45** | **684.09** | 94.91 |
| Steroid-M$_{lim}$ | Qwen-2.5 | 1750.99 | 901.99 | 106.99 | 1750.99 | 901.99 | 106.99 | 1750.99 | 901.99 | 106.99 | 1750.99 | 901.99 | 106.99 |
| | Mistral | 1737.63 | 896.17 | **95.80** | 1742.02 | 899.45 | 103.07 | 1738.92 | 898.42 | 103.42 | 1741.69 | 898.80 | 98.51 |
| | Falcon-3 | 1738.66 | 898.03 | 102.52 | 1740.75 | 898.42 | 98.74 | 1738.93 | 898.31 | 102.14 | 1742.69 | 900.19 | 97.78 |
| | LLaMA-2 | 1738.65 | 897.73 | 99.46 | 1741.24 | 898.86 | 100.51 | 1738.90 | 898.48 | 103.02 | 1743.15 | 900.89 | 102.91 |
| | LLaMA-3.1 | 1737.29 | 896.35 | 100.98 | 1741.25 | 899.54 | 100.01 | 1739.00 | 898.11 | 98.56 | 1743.21 | 900.77 | 98.70 |
| | Phi-4 | 1738.51 | 898.01 | 102.96 | 1741.42 | 898.43 | 97.71 | 1738.87 | 898.07 | 99.79 | 1743.05 | 900.66 | 98.94 |
| | DeepSeek-R1 | 1738.12 | 897.42 | 100.40 | 1740.81 | 898.72 | 98.67 | 1739.72 | 898.67 | 99.97 | 1743.62 | 900.59 | 98.15 |
| | **STT-LLM** | **1730.11** | **891.67** | **96.47** | **1733.18** | **893.01** | **96.47** | **1731.43** | **892.63** | **96.47** | **1734.87** | **894.61** | **96.47** |
| Steroid-F$_{lim}$ | Qwen-2.5 | 1309.99 | 666.99 | 127.99 | 1309.99 | 666.99 | 127.99 | 1309.99 | 666.99 | 127.99 | 1309.99 | 666.99 | 127.99 |
| | Mistral | 1292.73 | 657.49 | 123.29 | 1289.67 | 654.38 | 118.12 | 1294.05 | 657.05 | 107.06 | 1286.36 | 652.15 | 126.36 |
| | Falcon-3 | 1291.65 | 655.82 | 97.87 | 1289.63 | 654.51 | 96.85 | 1294.77 | 656.69 | 102.03 | 1287.89 | 653.04 | 96.47 |
| | LLaMA-2 | 1292.06 | 656.18 | 100.69 | 1289.05 | 653.63 | 93.96 | 1295.08 | 656.97 | 101.13 | 1287.68 | 654.31 | 106.19 |
| | LLaMA-3.1 | 1291.13 | 654.67 | **88.66** | 1289.66 | 654.13 | **93.20** | 1293.98 | 656.87 | 115.68 | 1287.32 | 653.76 | 108.22 |
| | Phi-4 | 1291.92 | 655.84 | 92.49 | 1289.48 | 654.17 | 97.54 | 1294.68 | 656.39 | 102.23 | 1287.37 | 653.85 | 107.41 |
| | DeepSeek-R1 | 1291.64 | 655.94 | 101.25 | 1289.33 | 654.37 | 103.85 | 1294.89 | 656.59 | 97.21 | 1286.90 | 653.18 | 101.71 |
| | **STT-LLM** | **1276.32** | **645.16** | 94.92 | **1274.23** | **643.71** | 94.89 | **1279.59** | **645.90** | **94.89** | **1272.19** | **642.90** | **94.86** |

confuse the model. The temporal tokenizer's contribution scales with sequence length: its removal drops sensitivity by 36% on Steroid-M (length 3-20) vs. 11% on Steroid-$F_{lim}$ (length 2), confirming increasing benefit for longer sequences. A small Transformer encoder produces embeddings but cannot generate natural language explanations nor support few-shot in-context learning. STT-LLM's advantage combines competitive prediction (Table 4) with contextual reasoning and monotonic few-shot improvement (Figure 3) within a single architecture.

Fig. 4 compares native LLM tokenization with STT-LLM across multiple general-purpose LLMs. Across all backbones, STT-LLM consistently improves anomaly sensitivity and reduces prediction error, showing that performance gains arise from the tokenization strategy rather than the choice of LLM architecture. By explicitly incorporating underlying metabolic pathway structure and temporal dependencies directly into the tokenization process, STT-LLM provides biologically grounded input representations that enable LLMs to better align with clinical dynamics.

## 7. Case Study

To evaluate the real-world applicability of our method, we conducted a case study on 29 longitudinal steroid profiles from real-world athletes, which were verified through DNA analysis by an anti-doping laboratory. Among these, 7 profiles were confirmed as anomalous due to different doping-related abnormalities, with domain experts providing detailed explanations, and the remaining 22 were classified as clean profiles. We used the clean profiles for sequence prediction and all 29 for anomaly detection. Our model achieved better forecasting performance with RMSE: 1673.13, MAE: 868.93, and MAPE: 95.51. For anomaly detection, the model accurately identified all 7 anomalous cases with 100% sensitivity, while misclassifying only 2 clean profiles (accuracy: 93.10%).

To evaluate the contextual reasoning ability of STT-LLM, we adopt a few-shot setup using the 7 expert-annotated longitudinal profiles to generate explanations for 500 additional profiles. These explanations were used to train all the models under identical training conditions. We then assessed model performance on the original 7 profiles (expert ground-truth explanations). As shown in Fig. 5, STT-LLM

*Table 3.* Local and global anomaly detection results across different datasets at zero-shot setting.

| Datasets | Baseline | Local | | | | | Global | | | | |
|---|---|---|---|---|---|---|---|---|---|---|---|
| | | Acc↑ | Sens↑ | Prec↑ | F1↑ | AUC↑ | Acc↑ | Sens↑ | Prec↑ | F1↑ | AUC↑ |
| Steroid-M | Qwen-2.5 | **0.96**±**.01** | 0.00±.00 | 0.00±.00 | 0.00±.00 | 0.47±.02 | 0.71±.02 | 0.08±.03 | 0.20±.02 | 0.11±.02 | 0.45±.02 |
| | Mistral | 0.87±.02 | 0.05±.01 | 0.02±.01 | 0.03±.01 | 0.43±.02 | 0.71±.02 | 0.08±.02 | 0.23±.03 | 0.12±.02 | 0.47±.02 |
| | Falcon-3 | 0.94±.02 | 0.01±.00 | 0.02±.01 | 0.01±.01 | 0.46±.02 | 0.72±.02 | 0.08±.02 | 0.28±.03 | 0.13±.02 | 0.53±.02 |
| | LLaMA-2 | 0.90±.02 | 0.05±.01 | 0.03±.01 | 0.04±.01 | 0.42±.02 | 0.71±.02 | 0.09±.02 | 0.26±.02 | 0.14±.03 | 0.49±.02 |
| | LLaMA-3.1 | 0.87±.02 | 0.07±.01 | 0.03±.01 | 0.05±.01 | 0.51±.02 | 0.72±.02 | 0.14±.02 | 0.33±.03 | 0.19±.03 | 0.56±.02 |
| | Phi-4 | 0.87±.02 | 0.08±.01 | 0.04±.01 | 0.05±.01 | 0.50±.02 | 0.72±.02 | 0.03±.01 | 0.15±.02 | 0.05±.01 | 0.46±.02 |
| | DeepSeek-R1 | 0.95±.01 | 0.02±.01 | 0.01±.01 | 0.01±.00 | 0.39±.02 | 0.70±.02 | 0.08±.02 | 0.21±.02 | 0.11±.02 | 0.45±.02 |
| | **STT-LLM** | 0.87±.02 | **0.15**±**.02** | **0.07**±**.01** | **0.09**±**.02** | **0.57**±**.02** | **0.73**±**.02** | **0.19**±**.03** | **0.41**±**.03** | **0.26**±**.03** | **0.57**±**.02** |
| Steroid-F | Qwen-2.5 | 0.87±.02 | 0.04±.01 | 0.02±.01 | 0.02±.01 | 0.46±.02 | 0.73±.02 | 0.04±.01 | 0.14±.02 | 0.06±.01 | 0.55±.02 |
| | Mistral | **0.96**±**.01** | 0.00±.00 | 0.00±.00 | 0.00±.00 | **0.62**±**.02** | 0.73±.02 | 0.12±.02 | 0.26±.03 | 0.16±.02 | 0.43±.02 |
| | Falcon-3 | 0.95±.01 | 0.00±.00 | 0.00±.00 | 0.00±.00 | 0.60±.02 | 0.72±.02 | 0.09±.02 | 0.22±.02 | 0.13±.02 | 0.37±.02 |
| | LLaMA-2 | 0.87±.02 | 0.06±.01 | 0.02±.01 | 0.03±.01 | 0.55±.02 | 0.73±.02 | 0.12±.02 | 0.26±.02 | 0.16±.02 | 0.47±.02 |
| | LLaMA-3.1 | 0.95±.01 | 0.01±.00 | 0.03±.01 | 0.02±.01 | 0.57±.02 | 0.73±.02 | 0.10±.02 | 0.23±.03 | 0.14±.02 | 0.49±.02 |
| | Phi-4 | 0.88±.02 | 0.06±.01 | 0.02±.01 | 0.03±.01 | 0.50±.02 | 0.74±.02 | 0.08±.02 | 0.25±.02 | 0.13±.02 | 0.45±.02 |
| | DeepSeek-R1 | 0.87±.02 | 0.06±.01 | 0.02±.01 | 0.03±.01 | 0.42±.02 | 0.73±.02 | 0.10±.02 | 0.22±.03 | 0.13±.02 | 0.50±.02 |
| | **STT-LLM** | 0.87±.02 | **0.08**±**.01** | **0.03**±**.01** | **0.05**±**.01** | 0.47±.02 | **0.75**±**.02** | **0.23**±**.03** | **0.40**±**.03** | **0.29**±**.03** | **0.59**±**.02** |
| Steroid-M$_{lim}$ | Qwen-2.5 | 0.86±.02 | 0.00±.00 | 0.00±.00 | 0.00±.00 | 0.18±.01 | 0.62±.02 | 0.06±.02 | 0.30±.03 | 0.10±.02 | 0.54±.02 |
| | Mistral | **0.96**±**.01** | 0.00±.00 | 0.00±.00 | 0.00±.00 | 0.37±.02 | 0.61±.02 | 0.07±.02 | 0.31±.02 | 0.12±.02 | 0.42±.02 |
| | Falcon-3 | 0.88±.02 | 0.08±.01 | 0.03±.01 | 0.05±.01 | 0.44±.02 | 0.61±.02 | 0.07±.02 | 0.32±.02 | 0.12±.02 | 0.53±.02 |
| | LLaMA-2 | 0.88±.02 | 0.03±.01 | 0.01±.01 | 0.02±.01 | 0.22±.01 | 0.61±.02 | 0.07±.02 | 0.31±.02 | 0.12±.02 | 0.45±.02 |
| | LLaMA-3.1 | 0.88±.02 | 0.04±.01 | 0.02±.01 | 0.02±.01 | 0.39±.02 | 0.60±.02 | 0.04±.01 | 0.21±.02 | 0.07±.02 | 0.39±.02 |
| | Phi-4 | 0.89±.02 | 0.21±.02 | 0.09±.01 | 0.12±.02 | 0.65±.02 | 0.61±.02 | 0.05±.01 | 0.25±.02 | 0.09±.01 | 0.44±.02 |
| | DeepSeek-R1 | 0.87±.02 | 0.06±.01 | 0.02±.01 | 0.03±.01 | 0.43±.02 | 0.60±.02 | 0.04±.01 | 0.19±.02 | 0.07±.01 | 0.45±.02 |
| | **STT-LLM** | 0.88±.02 | **0.36**±**.02** | **0.12**±**.02** | **0.18**±**.02** | **0.75**±**.02** | **0.64**±**.02** | **0.12**±**.02** | **0.47**±**.03** | **0.19**±**.02** | **0.55**±**.02** |
| Steroid-F$_{lim}$ | Qwen-2.5 | 0.88±.02 | 0.06±.01 | 0.06±.01 | 0.06±.01 | 0.14±.01 | 0.54±.02 | 0.10±.02 | 0.46±.03 | 0.16±.02 | 0.53±.02 |
| | Mistral | 0.95±.01 | 0.01±.00 | 0.03±.00 | 0.02±.00 | **0.64**±**.02** | 0.51±.02 | 0.04±.01 | 0.25±.02 | 0.07±.01 | 0.42±.02 |
| | Falcon-3 | **0.96**±**.01** | 0.00±.00 | 0.00±.00 | 0.00±.00 | 0.27±.02 | 0.55±.02 | 0.13±.02 | 0.53±.02 | 0.21±.03 | 0.55±.02 |
| | LLaMA-2 | 0.86±.02 | 0.03±.01 | 0.01±.01 | 0.02±.01 | 0.32±.02 | 0.54±.02 | 0.11±.02 | 0.48±.03 | 0.18±.02 | 0.50±.02 |
| | LLaMA-3.1 | 0.87±.02 | 0.00±.00 | 0.00±.00 | 0.00±.00 | 0.08±.01 | 0.52±.02 | 0.07±.01 | 0.36±.02 | 0.11±.01 | 0.46±.02 |
| | Phi-4 | 0.87±.02 | 0.07±.01 | 0.03±.01 | 0.04±.01 | 0.48±.02 | 0.53±.02 | 0.07±.01 | 0.41±.03 | 0.12±.02 | 0.48±.02 |
| | DeepSeek-R1 | 0.86±.02 | 0.10±.01 | 0.04±.01 | 0.06±.01 | 0.51±.02 | 0.54±.02 | 0.10±.02 | 0.48±.03 | 0.16±.02 | 0.54±.02 |
| | **STT-LLM** | 0.87±.02 | **0.17**±**.02** | **0.08**±**.01** | **0.11**±**.02** | 0.54±.02 | **0.59**±**.02** | **0.15**±**.03** | **0.71**±**.03** | **0.25**±**.03** | **0.56**±**.02** |

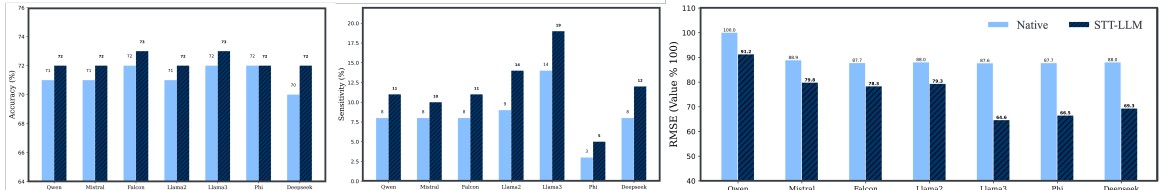

*Figure 4.* Performance comparison of LLM-native and STT-LLM tokenization strategies across different general-purpose LLMs on Steroid-M dataset.

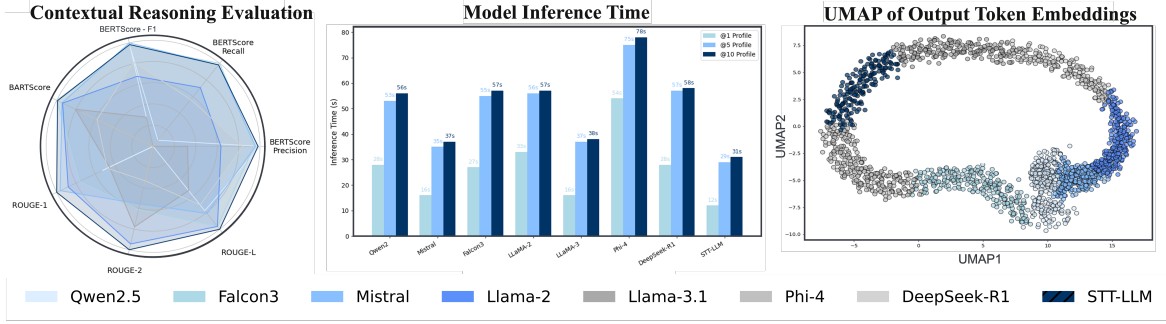

*Figure 5.* Evaluation of contextual reasoning quality (left), model inference time (center), and combined UMAP projection of output token embeddings from STT-LLM and LLM baselines (right).

*Table 4.* Performance of non-LLM baselines compared to STT-LLM on Steroid-M dataset. STT-LLM achieves competitive results against specialized architectures while additionally supporting contextual reasoning within a single unified pipeline.

| Non-LLM Baseline | Sequence Prediction | | | Anomaly Detection (Global) | | | | |
|---|---|---|---|---|---|---|---|---|
| | RMSE↓ | MAE↓ | MAPE↓ | Acc↑ | Sens↑ | Prec↑ | F1↑ | AUC↑ |
| TabPFN (Grinsztajn et al., 2025) | – | – | – | 0.5107 | 0.5032 | 0.0124 | 0.4019 | 0.4876 |
| MLP-SLAM (Li & Sun, 2024) | – | – | – | 0.4928 | 0.4215 | 0.0137 | 0.3284 | 0.4891 |
| M-GNN (Yuan & Nault, 2025) | – | – | – | 0.4716 | 0.4869 | 0.0043 | 0.6292 | 0.4738 |
| ARIMA (Schaffer et al., 2021) | 1675.47 | 879.96 | 96.99 | – | – | – | – | – |
| TDDGNN (Luo et al., 2024b) | 1675.78 | 880.61 | 98.91 | – | – | – | – | – |
| MAGNN (Chen et al., 2023) | 1675.70 | 880.44 | 98.29 | – | – | – | – | – |
| **STT-LLM** | **1664.59** | 881.20 | **96.80** | **0.7338** | 0.1935 | **0.4138** | 0.2637 | **0.5675** |

*Table 5.* Contributions of different components in STT-LLM on Steroid-M dataset.

| Model Variants | Sequence Prediction | | | Anomaly Detection (Global) | | | | |
|---|---|---|---|---|---|---|---|---|
| | RMSE↓ | MAE↓ | MAPE↓ | Acc↑ | Sens↑ | Prec↑ | F1↑ | AUC↑ |
| *w/o* all | 1687.71 | 896.39 | 98.93 | 0.7179 | 0.1398 | 0.3291 | 0.1962 | 0.5609 |
| *w/o* structural | 1687.49 | 896.61 | 100.65 | 0.7152 | 0.0968 | 0.2769 | 0.1434 | 0.4964 |
| *w/o* temporal | 1682.45 | 892.85 | 98.38 | 0.7126 | 0.1237 | 0.2987 | 0.1749 | 0.5500 |
| *w/o* embeddings | 1682.75 | 893.40 | 100.56 | 0.7139 | 0.1344 | 0.3125 | 0.1880 | 0.5352 |
| *w/o* structural + temporal | 1682.70 | 893.20 | 98.89 | 0.6967 | 0.0645 | 0.1791 | 0.0949 | 0.4877 |
| *w/o* embeddings + temporal | 1677.56 | 889.29 | 97.07 | 0.7245 | 0.1290 | 0.3429 | 0.1875 | 0.5474 |
| *w/o* embeddings + structural | 1679.16 | 891.78 | 97.35 | 0.7113 | 0.0914 | 0.2576 | 0.1349 | 0.4887 |
| **STT-LLM** | **1664.59** | **881.20** | **96.80** | **0.7338** | **0.1935** | **0.4138** | **0.2637** | **0.5675** |

outperforms all competing LLM baselines, showing higher alignment with expert interpretations. This highlights the model's ability to capture clinically meaningful reasoning patterns from limited supervision. In addition, we compared the inference efficiency of all models across different profile settings (1, 5, and 10 profiles). STT-LLM achieved substantially lower inference times than the baselines, requiring only 12s, 29s, and 31s respectively, whereas alternative LLMs ranged between 27-78s depending on model size and profile count. Finally, we visualize the output token embedding spaces of different models using UMAP representation. Unlike tightly clustered distributions, the embeddings form a continuous ring-like topology, suggesting a shared latent manifold, where STT-LLM occupies a transitional zone between LLaMA-3.1 and Phi-4. This placement suggests that STT-LLM maintains representational alignment with general-purpose LLMs while introducing localized structure unique to its domain-aware training.

## 8. Conclusion

We propose STT-LLM, a structural–temporal tokenization framework for adapting pretrained LLMs to longitudinal clinical data. By encoding temporal dependencies and pathway-induced structural constraints directly into LLM-compatible tokens, STT-LLM enables effective sequence prediction and anomaly detection under sparse and irregular sampling without modifying backbone architectures.

Empirical results on real-world sports anti-doping datasets demonstrate consistent gains over native LLM tokenization strategies. These findings position tokenization as a critical design dimension for extending general-purpose LLMs to longitudinal clinical domains.

**Limitations.** This work focuses on one-step-ahead prediction. Multi-step forecasting, where structural constraints may further reduce error accumulation, is left to future work.

## Acknowledgements

We thank the World Anti-Doping Agency (WADA) for supporting this work. We also thank the Reviewers and Program Chair for their constructive feedback.

## Impact Statement

This paper presents a methodological contribution aimed at advancing machine learning for longitudinal analysis. The proposed approach is intended to support expert analysis rather than automate decision-making and its potential applications align with established practices in clinical analytics.

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

# A. Technical Appendix

## A.1. Experimental Setup and Model Hyperparameters

**Model Configuration**    STT-LLM was trained using parameter-efficient fine-tuning via LoRA, where we systematically evaluated the impact of key hyperparameters on model performance. The final configuration uses a LoRA rank of 32, scaling factor (alpha) of 128, and a learning rate of 2e-5. To assess the sensitivity of the model to these choices, we conducted experiments on the Steroid-M dataset by varying one hyperparameter at a time while keeping others fixed. As shown in Table 6, reducing the LoRA rank to 16 led to a slight degradation in both sequence prediction (RMSE↑ +18.6) and anomaly detection (AUC↓ -0.006), while increasing the rank to 64 did not yield further gains. Similarly, modifying the alpha parameter to 64 or 256 degraded both predictive accuracy and detection precision, suggesting that 128 offers a balanced regularization. Finally, tuning the learning rate revealed that deviating from 2e-5, either lower (1e-6) or higher (2e-4) consistently reduced performance, particularly in terms of F1-score and AUC. These findings indicate that the selected configuration for STT-LLM strikes an optimal balance between predictive accuracy and detection sensitivity under constrained fine-tuning conditions. All experiments were conducted for 10 epochs using early stopping on a single NVIDIA Titan RTX GPU with 24GB memory.

**Projection Dimension and MLP Depth Sensitivity**    To assess the architectural design of our tokenizers, we study the impact of varying the projection dimension (PD) and the number of MLP layers in the structural and temporal tokenizers of STT-LLM. Our default configuration uses a projection dimension of 4096 and two MLP layers per tokenizer. As shown in Table 7, reducing the depth to a single MLP layer leads to a noticeable drop in detection performance, particularly sensitivity (-8.6%) and AUC (-0.046). Increasing the depth to three layers does not yield further gains, indicating that two layers strike a balance between expressivity and generalization. Similarly, projection dimensions of 1024 and 2048 underperform the 4096-dimensional variant, especially on precision and F1-score. The model variant with 4096 PD and 2-layer MLPs achieves the highest performance across all anomaly detection metrics (e.g., AUC: 0.5675, F1: 0.2637), highlighting the importance of sufficient projection capacity and moderate depth in capturing clinically relevant temporal and structural patterns.

**Prompt Design for Task-Specific Supervision**    To enable task-specific training and evaluation across all models, including STT-LLM, we designed structured natural language prompts tailored to three core objectives: reasoning, classification, and sequence prediction. As illustrated in Fig. 6, the reasoning prompt asks the model to detect and explain abnormalities within a given steroid profile across multiple time points, encouraging contextual understanding and interpretability. The classification prompt explicitly instructs the model to either confirm the consistency of normal profiles or identify and explain the presence of an anomaly in the final sample. In contrast, the prediction prompt emphasizes learning temporal patterns, either under the assumption of normality or while acknowledging that the last sample deviates from the expected trend. These prompts allow all models to operate in a unified few-shot setting while supporting gradient-based fine-tuning or embedding-level supervision, depending on the architecture. They also ensure that the models are evaluated consistently across both descriptive and diagnostic clinical tasks.

## A.2. Detailed Results

### A.2.1. Model Loss Performance

We report the training and evaluation loss curves for both sequence prediction and anomaly detection tasks over 10 training epochs in Fig. 7. For sequence prediction (left), the model shows rapid convergence, with the training loss decreasing sharply within the first 5 epochs and plateauing around 620. The evaluation loss follows a similar trend, stabilizing around epoch 5, indicating strong generalization without overfitting. For anomaly detection (right), both training and evaluation losses exhibit an even faster convergence, with steep declines in the first 3 epochs and near-flattening thereafter around a value of 0.2. This consistency between train and eval curves in both tasks demonstrates that the STT-LLM framework effectively learns stable representations with minimal overfitting, even in low-resource settings. The rapid convergence further underscores the efficiency of the proposed structural-temporal tokenization strategy, which enables fast adaptation to downstream clinical tasks with minimal tuning.

*Table 6.* Contributions of different hyperparameter configurations in STT-LLM on Steroid-M dataset.

| Model Variants | Sequence Prediction | | | Anomaly Detection (Global) | | | | |
|---|---|---|---|---|---|---|---|---|
| | RMSE↓ | MAE↓ | MAPE↓ | Acc↑ | Sens↑ | Prec↑ | F1↑ | AUC↑ |
| Lower Rank (16) | 1683.22 | 893.40 | 97.96 | 0.7179 | 0.1398 | 0.3291 | 0.1962 | 0.5609 |
| Higher Rank (64) | 1687.49 | 896.61 | 100.65 | 0.7152 | 0.0968 | 0.2769 | 0.1434 | 0.4964 |
| Lower alpha (64) | 1668.54 | 883.90 | 100.00 | 0.7126 | 0.1237 | 0.2987 | 0.1749 | 0.5500 |
| Higher alpha (256) | 1682.75 | 893.40 | 100.56 | 0.7139 | 0.1344 | 0.3125 | 0.1880 | 0.5352 |
| Lower LR (1e-6) | 1679.46 | 890.80 | 98.64 | 0.7245 | 0.0645 | 0.2609 | 0.1034 | 0.5034 |
| Higher LR (2e-4) | 1689.05 | 897.55 | 100.23 | 0.7126 | 0.1183 | 0.2933 | 0.1686 | 0.5005 |
| **STT-LLM** | **1664.59** | **881.20** | **96.80** | **0.7338** | **0.1935** | **0.4138** | **0.2637** | **0.5675** |

*Table 7.* Contributions of different Projection Dimensions (PD) and MLP layers in STT-LLM on Steroid-M dataset.

| Model Variants | Anomaly Detection (Global) | | | | |
|---|---|---|---|---|---|
| | Acc↑ | Sens↑ | Prec↑ | F1↑ | AUC↑ |
| 1 MLP layer | 0.7060 | 0.1075 | 0.2632 | 0.1527 | 0.5214 |
| 3 MLP layers | 0.7086 | 0.1183 | 0.2821 | 0.1667 | 0.5103 |
| 1024 PD | 0.7232 | 0.0753 | 0.2745 | 0.1181 | 0.5026 |
| 2048 PD | 0.7285 | 0.1129 | 0.3443 | 0.1700 | 0.5490 |
| **STT-LLM** | **0.7338** | **0.1935** | **0.4138** | **0.2637** | **0.5675** |

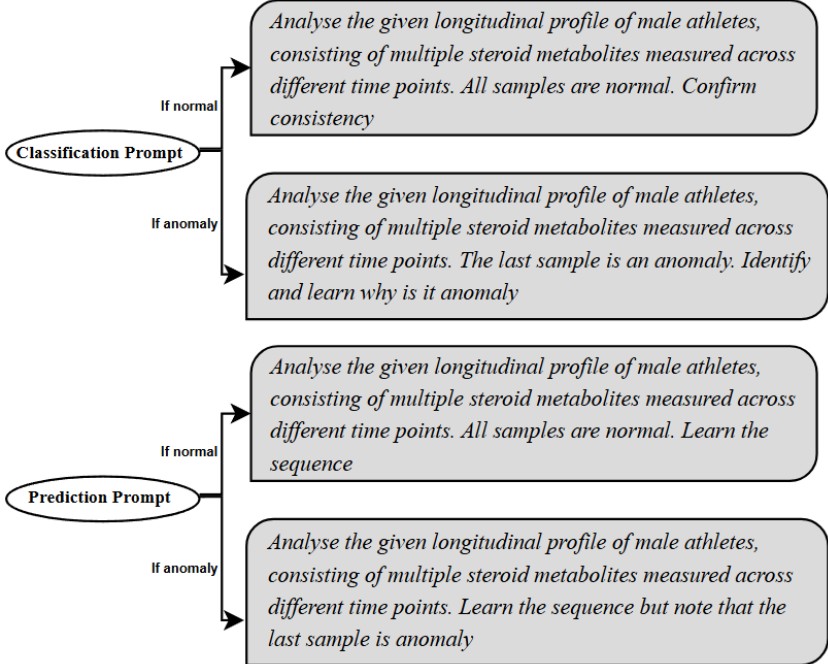

*Figure 6.* Prompts for training STT-LLM for different tasks.

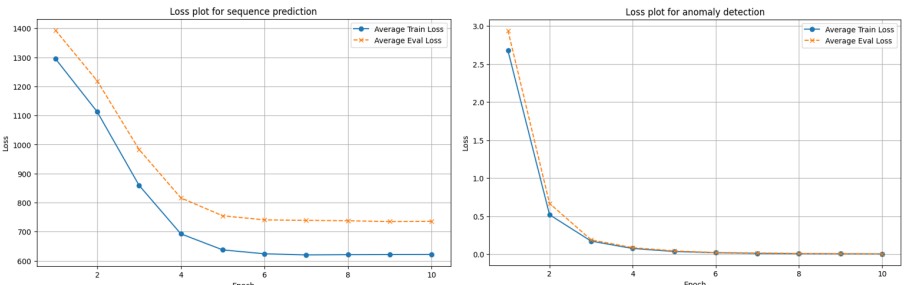

*Figure 7.* Loss plots for sequence prediction & anomaly detection tasks.

### A.2.2. SEQUENCE PREDICTION

**Few-shot settings**  As shown in Fig. 8 and Table 8, STT-LLM consistently outperforms all competing LLM baselines across all datasets and shot configurations in terms of RMSE, MAE, and MAPE. STT-LLM maintains stable and low error margins across shot variations, while other models (Qwen-2.5 and Mistral) show higher variance and sensitivity to shot count. On the Steroid-M dataset, STT-LLM achieves the best RMSE (1664.59), MAE (881.20), and MAPE (96.80) in the 2-shot setting and retains this advantage throughout. This performance trend generalizes to the limited-data settings (Steroid-$M_{lim}$, Steroid-$F_{lim}$), where the tokenization-aware STT-LLM demonstrates stronger robustness and lower generalization error. These results validate the model's ability to learn meaningful temporal patterns under constrained supervision and emphasize the effectiveness of its structural-temporal embedding design for few-shot sequence modeling in longitudinal clinical data.

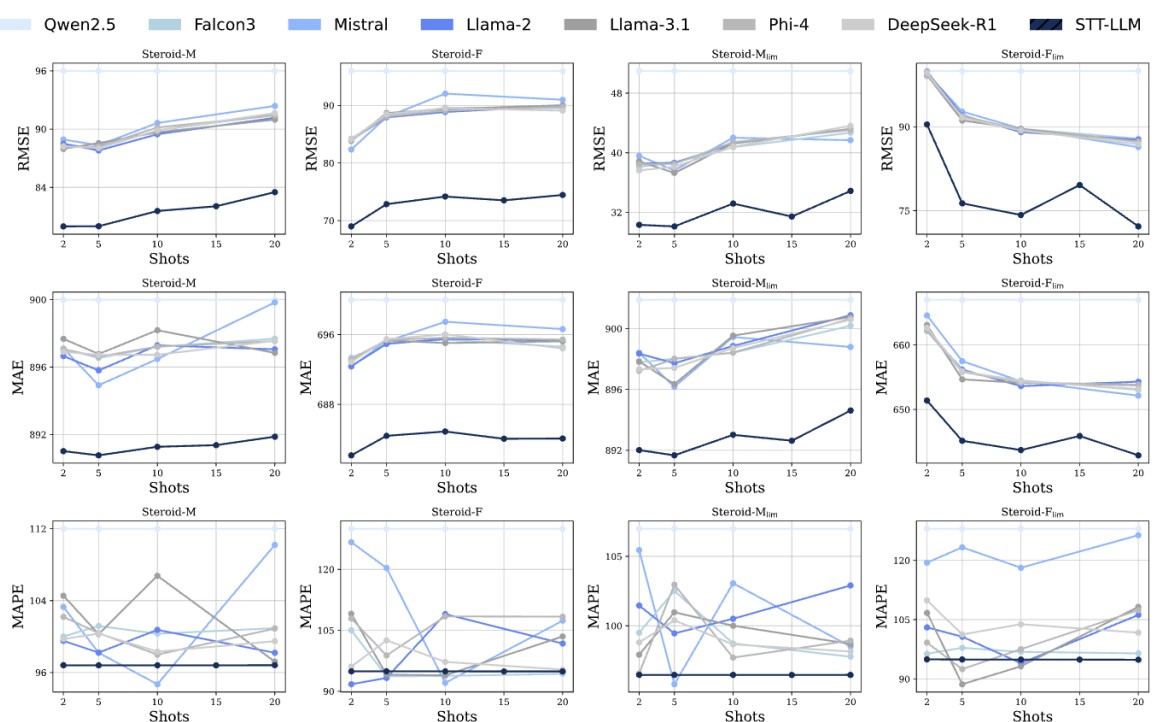

*Figure 8.* Few-shot learning results for sequence prediction.

### A.2.3. ANOMALY DETECTION

**Zero-Shot Local Anomaly Detection**  Fig. 9 presents the zero-shot local anomaly detection performance across four datasets. STT-LLM consistently outperforms all baseline models across all metrics. It achieves the highest sensitivity on Steroid-M and Steroid-$M_{lim}$ (16.8% and 32.4%, respectively), which are particularly challenging due to subtle temporal deviations. Additionally, STT-LLM shows significantly higher F1-scores (up to 18.4%) and precision values compared

*Table 8.* Few-shot (2, 5, 10, 15, 20) sequence prediction results across different datasets.

| Datasets | Model | @2 | | | @5 | | | @10 | | | @15 | | | @20 | | |
|---|---|---|---|---|---|---|---|---|---|---|---|---|---|---|---|---|
| | | RMSE↓ | MAE↓ | MAPE↓ | RMSE↓ | MAE↓ | MAPE↓ | RMSE↓ | MAE↓ | MAPE↓ | RMSE↓ | MAE↓ | MAPE↓ | RMSE↓ | MAE↓ | MAPE↓ |
| Steroid-M | Qwen-2.5 | 1695.99 | 899.99 | 111.99 | 1695.99 | 899.99 | 111.99 | 1695.99 | 899.99 | 111.99 | 1695.99 | 899.99 | 111.99 | 1695.99 | 899.99 | 111.99 |
| | Mistral | 1688.93 | 897.11 | 103.29 | 1688.34 | 894.92 | 98.19 | 1690.63 | 896.48 | **94.69** | 1688.90 | 896.54 | 101.04 | 1692.39 | 899.84 | 110.19 |
| | Falcon-3 | 1688.43 | 897.08 | 100.00 | 1688.02 | 896.54 | 101.17 | 1689.88 | 897.20 | 100.31 | 1690.39 | 897.48 | 100.01 | 1691.48 | 897.69 | 100.93 |
| | LLaMA-2 | 1688.50 | 896.65 | 99.51 | 1687.80 | 895.81 | 98.21 | 1689.47 | 897.29 | 100.74 | 1690.01 | 896.86 | 100.47 | 1691.16 | 897.06 | 98.19 |
| | LLaMA-3.1 | 1687.96 | 897.67 | 104.53 | 1688.57 | 896.74 | 100.27 | 1689.67 | 898.19 | 106.75 | 1690.56 | 897.17 | 101.46 | 1690.98 | 896.84 | 97.21 |
| | Phi-4 | 1688.14 | 897.07 | 102.19 | 1688.20 | 896.62 | 100.38 | 1690.17 | 897.21 | 97.98 | 1690.04 | 897.36 | 102.87 | 1691.41 | 897.54 | 100.88 |
| | DeepSeek-R1 | 1688.25 | 896.91 | 99.72 | 1688.05 | 896.73 | 100.33 | 1689.88 | 896.73 | 98.31 | 1690.31 | 897.17 | 98.81 | 1691.65 | 897.56 | 99.48 |
| | **STT-LLM** | **1679.99** | **891.02** | **96.80** | **1680.00** | **890.77** | **96.80** | **1681.57** | **891.27** | 96.79 | **1682.06** | **891.37** | 96.79 | **1683.51** | **891.87** | 96.81 |
| Steroid-F | Qwen-2.5 | 1395.99 | 699.99 | 129.99 | 1395.99 | 699.99 | 129.99 | 1395.99 | 699.99 | 129.99 | 1395.99 | 699.99 | 129.99 | 1395.99 | 699.99 | 129.99 |
| | Mistral | 1382.34 | 692.33 | 126.69 | 1387.98 | 695.23 | 120.35 | 1392.05 | 697.48 | **92.08** | 1388.75 | 695.68 | 108.68 | 1390.98 | 696.63 | 107.37 |
| | Falcon-3 | 1384.25 | 693.37 | 105.03 | 1388.12 | 695.07 | 93.71 | 1389.62 | 695.41 | 93.80 | 1388.77 | 695.67 | 115.24 | 1389.53 | 694.61 | **94.31** |
| | LLaMA-2 | 1384.00 | 692.38 | **91.75** | 1387.93 | 694.93 | 93.30 | 1388.86 | 695.52 | 109.01 | 1388.92 | 694.91 | 100.61 | 1389.93 | 695.33 | 101.76 |
| | LLaMA-3.1 | 1383.84 | 693.10 | 109.09 | 1388.67 | 695.34 | 94.12 | 1389.38 | 695.03 | 93.93 | 1388.72 | 695.19 | 106.55 | 1390.03 | 695.23 | 103.50 |
| | Phi-4 | 1383.99 | 693.18 | 107.90 | 1388.07 | 695.39 | 98.83 | 1389.09 | 695.64 | 108.44 | 1389.50 | 694.70 | 99.72 | 1389.75 | 695.43 | 108.37 |
| | DeepSeek-R1 | 1384.11 | 692.87 | 96.08 | 1388.48 | 695.47 | 102.55 | 1389.54 | 696.04 | 97.25 | 1389.05 | 695.14 | 98.93 | 1389.09 | 694.41 | 95.36 |
| | **STT-LLM** | **1368.99** | **682.16** | 94.92 | **1372.85** | **684.39** | 94.94 | **1374.17** | **684.89** | 94.92 | **1373.51** | **684.05** | 94.91 | **1374.45** | **684.09** | 94.91 |
| Steroid-M$_{lim}$ | Qwen-2.5 | 1750.99 | 901.99 | 106.99 | 1750.99 | 901.99 | 106.99 | 1750.99 | 901.99 | 106.99 | 1750.99 | 901.99 | 106.99 | 1750.99 | 901.99 | 106.99 |
| | Mistral | 1739.59 | 898.42 | 105.45 | 1737.63 | 896.17 | **95.80** | 1742.02 | 899.45 | 103.07 | 1738.92 | 898.42 | 103.42 | 1741.69 | 898.80 | 98.51 |
| | Falcon-3 | 1738.11 | 897.78 | 99.51 | 1738.66 | 898.03 | 102.52 | 1740.75 | 898.42 | 98.74 | 1738.93 | 898.31 | 102.14 | 1742.69 | 900.19 | 97.78 |
| | LLaMA-2 | 1738.57 | 898.38 | 101.46 | 1738.65 | 897.73 | 99.46 | 1741.24 | 898.86 | 100.51 | 1738.90 | 898.48 | 103.02 | 1743.15 | 900.89 | 102.91 |
| | LLaMA-3.1 | 1738.86 | 897.85 | 97.92 | 1738.27 | 896.35 | 100.98 | 1741.25 | 899.54 | 100.01 | 1739.00 | 898.15 | 98.56 | 1743.21 | 900.77 | 98.70 |
| | Phi-4 | 1738.46 | 897.22 | 96.57 | 1738.51 | 898.01 | 102.96 | 1741.42 | 898.43 | 97.71 | 1738.87 | 898.07 | 99.79 | 1743.05 | 900.66 | 98.94 |
| | DeepSeek-R1 | 1737.63 | 897.33 | 98.81 | 1738.12 | 897.42 | 100.40 | 1740.81 | 898.72 | 98.67 | 1739.72 | 898.67 | 99.97 | 1743.62 | 900.59 | 98.15 |
| | **STT-LLM** | **1730.32** | **892.01** | **96.47** | **1730.11** | **891.67** | **96.47** | **1733.18** | **893.01** | **96.47** | **1731.43** | **892.63** | **96.47** | **1734.87** | **894.61** | **96.47** |
| Steroid-F$_{lim}$ | Qwen-2.5 | 1309.99 | 666.99 | 127.99 | 1309.99 | 666.99 | 127.99 | 1309.99 | 666.99 | 127.99 | 1309.99 | 666.99 | 127.99 | 1309.99 | 666.99 | 127.99 |
| | Mistral | 1307.58 | 664.58 | 119.41 | 1292.73 | 657.49 | 123.29 | 1289.67 | 652.95 | 127.06 | 1294.05 | 657.05 | 107.06 | 1286.36 | 652.15 | 126.36 |
| | Falcon-3 | 1305.97 | 662.14 | 96.36 | 1291.65 | 655.82 | 97.87 | 1289.63 | 654.51 | 96.85 | 1294.77 | 656.69 | 102.03 | 1287.89 | 653.04 | 96.47 |
| | LLaMA-2 | 1305.82 | 662.30 | 103.04 | 1292.06 | 656.18 | 100.69 | 1289.05 | 653.63 | 93.96 | 1295.08 | 656.97 | 101.13 | 1287.68 | 654.31 | 106.19 |
| | LLaMA-3.1 | 1306.20 | 663.09 | 106.72 | 1291.13 | 654.67 | **88.66** | 1289.66 | 654.13 | 93.20 | 1293.98 | 656.87 | 115.68 | 1287.32 | 653.76 | 108.22 |
| | Phi-4 | 1306.07 | 662.29 | 99.20 | 1291.92 | 655.84 | 92.49 | 1289.48 | 654.17 | 97.54 | 1294.68 | 656.39 | 102.23 | 1287.37 | 653.85 | 107.41 |
| | DeepSeek-R1 | 1305.70 | 662.66 | 109.88 | 1291.64 | 655.94 | 101.25 | 1289.33 | 654.37 | 103.85 | 1294.89 | 656.59 | 97.21 | 1286.90 | 653.18 | 101.71 |
| | **STT-LLM** | **1290.41** | **651.38** | 94.95 | **1276.32** | **645.16** | 94.92 | **1274.23** | **643.71** | 94.89 | **1279.59** | **645.90** | 94.89 | **1272.19** | **642.90** | 94.86 |

to all baselines, indicating its capacity to correctly identify rare anomalous samples with minimal false positives. These results underscore the effectiveness of the structural-temporal tokenization in capturing fine-grained temporal inconsistencies without additional task-specific fine-tuning.

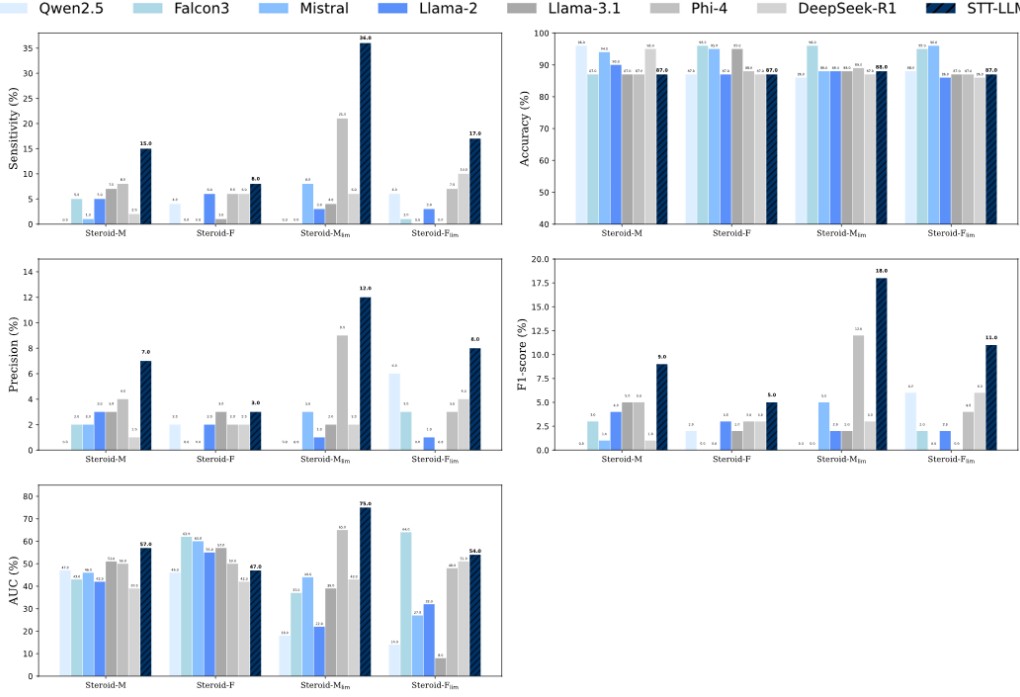

*Figure 9.* Zero-shot local anomaly detection performance across different metrics.

**Few-Shot Local Anomaly Detection**   Fig. 10 shows the few-shot learning performance of all models on local anomaly detection across 2, 5, 10, 15, and 20-shot configurations. STT-LLM demonstrates strong adaptability, achieving the highest or near-highest scores across most metrics and datasets, particularly under 5- and 10-shot settings. Unlike many baselines that fluctuate substantially across shots, STT-LLM maintains stable upward trends in precision, sensitivity, and AUC. For example, in the Steroid-$F_{lim}$ dataset, STT-LLM shows steady improvements in both F1-score and AUC, highlighting its robustness in low-resource regimes. The combination of structural and temporal embeddings appears to improve its generalization in clinical settings with sparse anomaly labels.

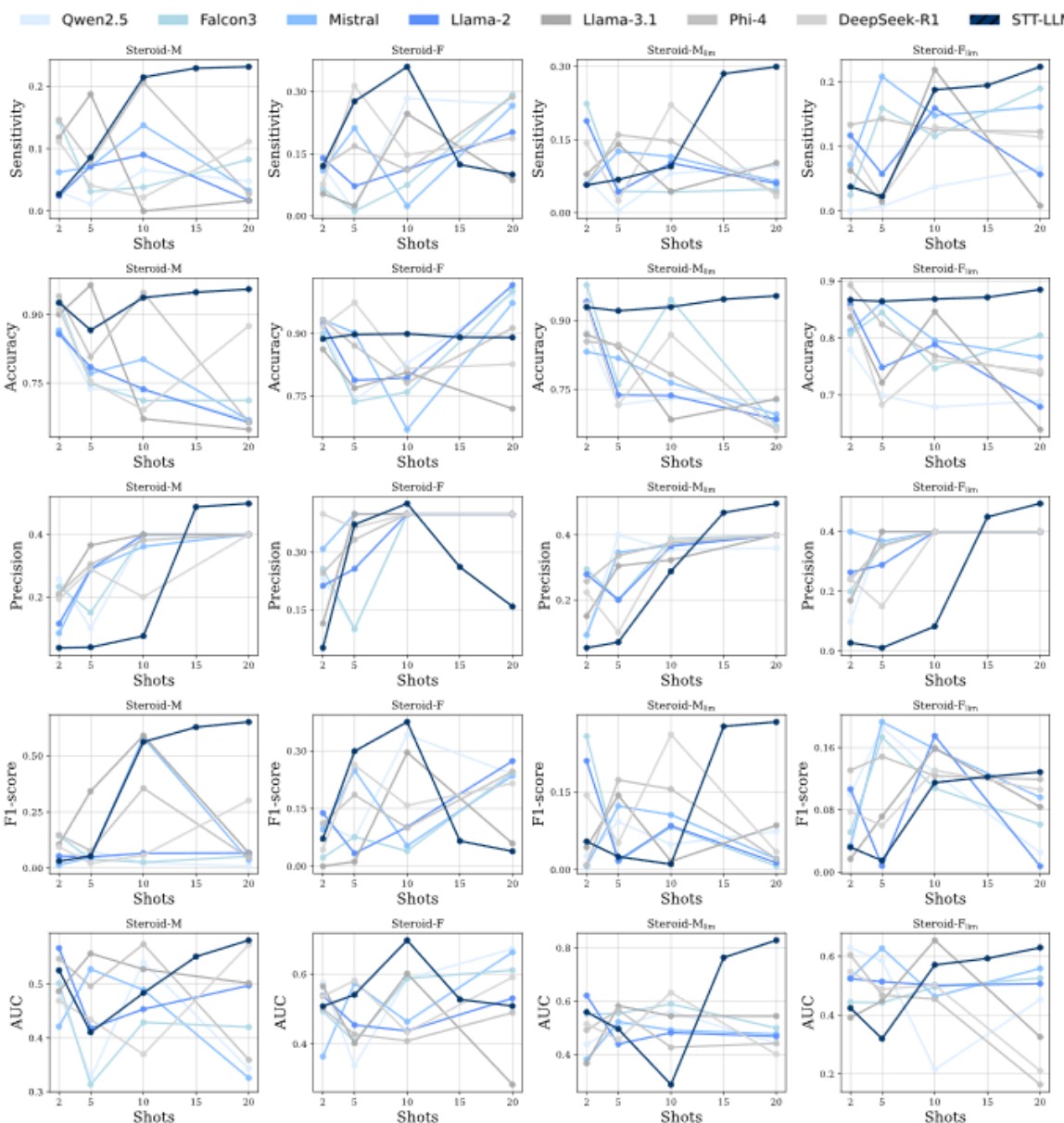

*Figure 10.* Few-shot local anomaly detection performance across different metrics.

**Zero-Shot Global Anomaly Detection**   In Fig. 11, we shows global anomaly detection performance under zero-shot evaluation. STT-LLM outperforms all baselines across most metrics and datasets, achieving the highest F1-scores and AUC on all datasets, including challenging low-data subsets like Steroid-$F_{lim}$. For example, it achieves 26.8% F1-score and 78.6% AUC on Steroid-$F_{lim}$, significantly outperforming the second-best model. Moreover, STT-LLM maintains a strong balance

across precision and sensitivity, indicating its ability to detect true anomalies without overfitting to normal patterns. This demonstrates that STT-LLM can generalize effectively even when provided with no additional in-context examples.

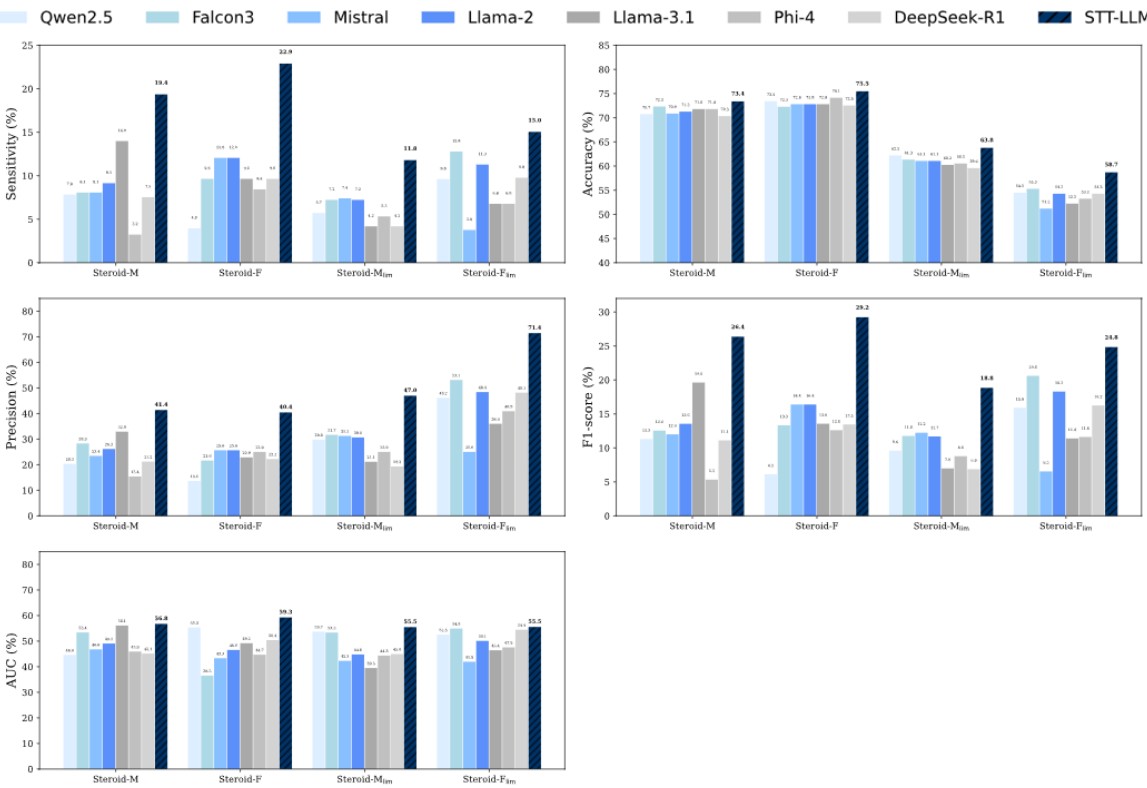

*Figure 11.* Zero-shot global anomaly detection performance across different metrics.

**Few-Shot Global Anomaly Detection**   Fig. 12 shows the model performance under few-shot global anomaly detection. STT-LLM not only maintains competitive performance in low-shot scenarios but also scales more effectively with additional context compared to baselines. It consistently leads in AUC and F1-score across 10- and 20-shot settings, especially on Steroid-M and Steroid-F datasets. Unlike other models that show unstable or non-monotonic performance trends, STT-LLM improves predictably with more shots, demonstrating strong in-context learning capabilities for rare-event detection. This highlights the strength of its tokenization scheme in enabling efficient information transfer even with minimal labeled supervision.

### A.3. Ablation Studies

To understand the individual contributions of STT-LLM's components, we perform extensive ablation studies across all four datasets. As shown in Tables 9-12, the STT-LLM configuration achieves the best performance across nearly all metrics for both sequence prediction and anomaly detection. When the structural tokenizer is removed, there is a consistent drop in AUC (e.g., from 0.5675 to 0.4964 on Steroid-M, and 0.5927 to 0.5090 on Steroid-F), suggesting that capturing biochemical dependencies between steroid metabolites is important for anomaly discrimination. Similarly, ablating the temporal tokenizer leads to substantial reductions in sensitivity and F1-score, particularly in clinically relevant low-data conditions (e.g., Steroid-$F_{lim}$: F1 drops from 0.2484 to 0.0559).

The projection embedding layer, which aligns structural-temporal features to the LLM-compatible token space, also proves essential. On Steroid-$M_{lim}$, removing embeddings causes the largest AUC drop (0.5548 to 0.5458), and precision degrades across all datasets. Combinations of missing components further compound performance loss. For example, removing both structural and temporal components results in the weakest performance in nearly every metric (e.g., AUC of 0.4877 on Steroid-M and 0.4353 on Steroid-$F_{lim}$). These degradations indicate that no single component is independently sufficient; rather, their integration is key to capturing both temporal variation and physiological structure in a format usable by frozen

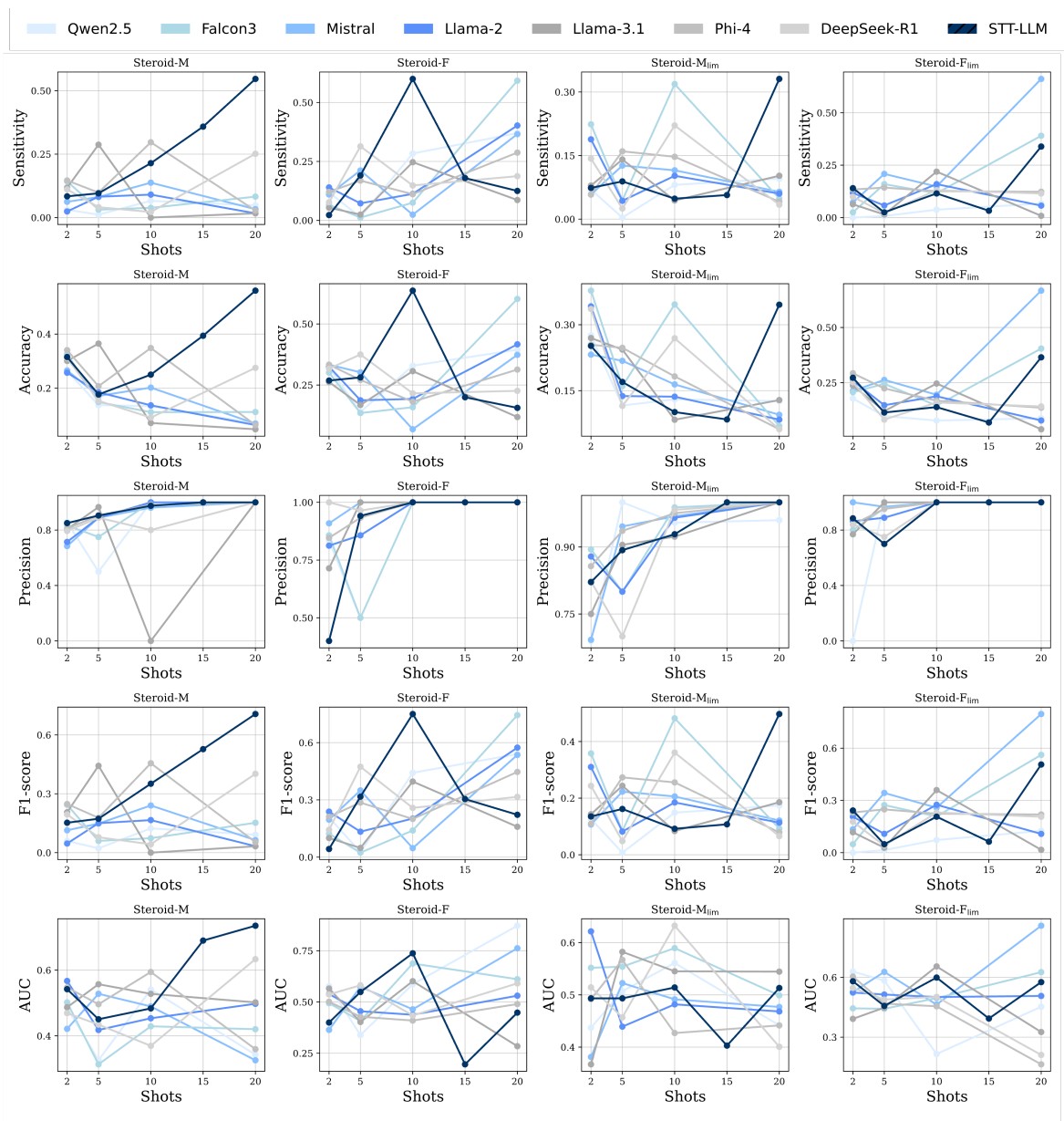

*Figure 12.* Few-shot global anomaly detection performance across different metrics.

LLMs. Overall, these ablation results affirm the architectural design of STT-LLM. The synergy between the structural tokenizer (capturing inter-variable relations), temporal tokenizer (capturing temporal evolution), and embedding projection (aligning with LLM input semantics) is important for robust generalization. The consistency of findings across diverse datasets and varying data availability further supports the adaptability and modular design benefits of STT-LLM in real-world clinical anomaly detection and forecasting tasks.

### A.4. Per-Backbone STT-LLM Performance Gains

To verify that STT-LLM's improvements are not specific to its default LLaMA-3.1 (8B) backbone, Table 13 reports the sensitivity gain (anomaly detection, in percentage points) and RMSE reduction (sequence prediction, %) achieved by STT-LLM tokenization over native tokenization for each of the seven backbone LLMs evaluated on the STEROID-M dataset. Gains are consistent across all backbones, confirming that the performance improvements arise from the structural-temporal tokenization framework rather than the backbone architecture.

*Table 9.* Contributions of different components in STT-LLM on Steroid-M.

| Model Variants | Sequence Prediction | | | Anomaly Detection (Global) | | | | |
|---|---|---|---|---|---|---|---|---|
| | RMSE↓ | MAE↓ | MAPE↓ | Acc↑ | Sens↑ | Prec↑ | F1↑ | AUC↑ |
| *w/o* all | 1687.71 | 896.39 | 98.93 | 0.7179 | 0.1398 | 0.3291 | 0.1962 | 0.5609 |
| *w/o* structural | 1687.49 | 896.61 | 100.65 | 0.7152 | 0.0968 | 0.2769 | 0.1434 | 0.4964 |
| *w/o* temporal | 1682.45 | 892.85 | 98.38 | 0.7126 | 0.1237 | 0.2987 | 0.1749 | 0.5500 |
| *w/o* embeddings | 1682.75 | 893.40 | 100.56 | 0.7139 | 0.1344 | 0.3125 | 0.1880 | 0.5352 |
| *w/o* structural + temporal | 1682.70 | 893.20 | 98.89 | 0.6967 | 0.0645 | 0.1791 | 0.0949 | 0.4877 |
| *w/o* embeddings + temporal | 1677.56 | 889.29 | 97.07 | 0.7245 | 0.1290 | 0.3429 | 0.1875 | 0.5474 |
| *w/o* embeddings + structural | 1679.16 | 891.78 | 97.35 | 0.7113 | 0.0914 | 0.2576 | 0.1349 | 0.4887 |
| **STT-LLM** | **1664.59** | **881.20** | **96.80** | **0.7338** | **0.1935** | **0.4138** | **0.2637** | **0.5675** |

*Table 10.* Contributions of different components in STT-LLM on Steroid-F.

| Model Variants | Sequence Prediction | | | Anomaly Detection (Global) | | | | |
|---|---|---|---|---|---|---|---|---|
| | RMSE↓ | MAE↓ | MAPE↓ | Acc↑ | Sens↑ | Prec↑ | F1↑ | AUC↑ |
| *w/o* all | 1384.26 | 693.99 | 114.88 | 0.7280 | 0.0964 | 0.2286 | 0.1356 | 0.4919 |
| *w/o* structural | 1368.79 | 682.52 | 97.64 | 0.7467 | 0.1687 | 0.3500 | 0.2276 | 0.5090 |
| *w/o* temporal | 1369.23 | 683.17 | 101.36 | 0.7280 | 0.0620 | 0.1724 | 0.0893 | 0.4682 |
| *w/o* embeddings | 1383.83 | 693.49 | 104.61 | 0.7413 | 0.1807 | 0.3409 | 0.2362 | 0.4998 |
| *w/o* structural + temporal | 1373.63 | 686.26 | 99.16 | 0.7307 | 0.0482 | 0.1538 | 0.0734 | 0.4017 |
| *w/o* embeddings + temporal | 1380.30 | 690.67 | 95.46 | 0.7493 | 0.1928 | 0.3721 | 0.2540 | 0.5909 |
| *w/o* embeddings + structural | 1384.13 | 693.57 | 100.04 | 0.7333 | 0.1205 | 0.2703 | 0.1667 | 0.4663 |
| **STT-LLM** | **1368.44** | **682.39** | **94.92** | **0.7547** | **0.2289** | **0.4043** | **0.2923** | **0.5927** |

*Table 11.* Contributions of different components in STT-LLM on Steroid-M$_{\text{lim}}$.

| Model Variants | Sequence Prediction | | | Anomaly Detection (Global) | | | | |
|---|---|---|---|---|---|---|---|---|
| | RMSE↓ | MAE↓ | MAPE↓ | Acc↑ | Sens↑ | Prec↑ | F1↑ | AUC↑ |
| *w/o* all | 1737.36 | 897.95 | 102.07 | 0.6024 | 0.0418 | 0.2115 | 0.0698 | 0.3949 |
| *w/o* structural | 1737.92 | 897.88 | 100.73 | 0.6214 | 0.0570 | 0.3261 | 0.0971 | 0.5109 |
| *w/o* temporal | 1732.97 | 894.13 | 98.19 | 0.6174 | 0.0913 | 0.3582 | 0.1455 | 0.4199 |
| *w/o* embeddings | 1733.26 | 894.47 | 98.78 | 0.6269 | 0.0993 | 0.4250 | 0.1783 | 0.5458 |
| *w/o* structural + temporal | 1733.22 | 893.20 | 98.89 | 0.6159 | 0.0645 | 0.1791 | 0.0949 | 0.4877 |
| *w/o* embeddings + temporal | 1732.17 | 892.56 | 96.78 | 0.6364 | 0.0038 | 0.1429 | 0.0074 | 0.3759 |
| *w/o* embeddings + structural | 1731.42 | 892.27 | 97.09 | 0.6119 | 0.0608 | 0.2909 | 0.1006 | 0.4373 |
| **STT-LLM** | **1730.04** | **892.08** | **96.46** | **0.6377** | **0.1179** | **0.4697** | **0.1884** | **0.5548** |

*Table 12.* Contributions of different components in STT-LLM on Steroid-F$_{\text{lim}}$.

| Model Variants | Sequence Prediction | | | Anomaly Detection (Global) | | | | |
|---|---|---|---|---|---|---|---|---|
| | RMSE↓ | MAE↓ | MAPE↓ | Acc↑ | Sens↑ | Prec↑ | F1↑ | AUC↑ |
| *w/o* all | 1305.98 | 662.99 | 100.47 | 0.5222 | 0.0677 | 0.3600 | 0.1139 | 0.4644 |
| *w/o* structural | 1301.38 | 651.91 | 97.40 | 0.5085 | 0.0376 | 0.2381 | 0.0649 | 0.4056 |
| *w/o* temporal | 1301.44 | 652.57 | 101.23 | 0.5392 | 0.0301 | 0.4000 | 0.0559 | 0.4946 |
| *w/o* embeddings | 1305.87 | 662.88 | 104.29 | 0.5392 | 0.0977 | 0.4643 | 0.1615 | 0.5090 |
| *w/o* structural + temporal | 1301.52 | 655.66 | 99.21 | 0.5222 | 0.0226 | 0.2308 | 0.0411 | 0.4353 |
| *w/o* embeddings + temporal | 1302.30 | 660.06 | 95.61 | 0.5324 | 0.0902 | 0.4286 | 0.1491 | 0.4128 |
| *w/o* embeddings + structural | 1306.18 | 662.97 | 100.18 | 0.5392 | 0.0301 | 0.4000 | 0.0559 | 0.4792 |
| **STT-LLM** | **1301.24** | **651.79** | **94.97** | **0.5870** | **0.1504** | **0.7143** | **0.2484** | **0.5555** |

*Table 13.* Per-backbone sensitivity gain (pp) and RMSE reduction (%) of STT-LLM over native tokenization on STEROID-M.

| Backbone | Sensitivity Gain (pp) | RMSE Reduction (%) |
|---|---|---|
| Qwen-2.5 | +3 | 9.35 |
| Mistral | +2 | 10.24 |
| Falcon-3 | +3 | 10.72 |
| LLaMA-2 | +5 | 9.89 |
| LLaMA-3.1 | +14 | 26.26 |
| Phi-4 | +5 | 24.17 |
| DeepSeek-R1 | +4 | 19.42 |

## A.5. Case Study

To evaluate the contextual reasoning ability of STT-LLM, we conducted a controlled comparison against several strong baseline LLMs using a carefully designed prompt. The prompt simulates a real-world scenario where a model should analyze a longitudinal steroid profile consisting of six metabolites measured across multiple time points and answer three questions: (1) identify the anomalous sample, (2) provide a reason based on the steroid metabolism pathway, and (3) determine whether the anomaly is due to steroid doping or a mismatched sample. Fig. 13 shows the generated responses of different LLMs on a profile with a known anomaly (Sample 4) caused by steroid doping.

As shown in the responses, most LLMs either failed to detect the correct anomaly or provided vague and incoherent explanations. For example, Phi and Qwen misattributed the anomaly to sample swapping despite correctly identifying the anomalous sample. DeepSeek-R1 failed to reach a decisive conclusion, and Falcon made inconsistent inferences based on isolated metabolite values. Although Mistral and LLaMA-3 correctly identified Sample 4, their reasoning relied on static domain knowledge (e.g., typical metabolite levels) rather than relational comparison across samples in the profile. This limitation arises because general-purpose LLMs are not inherently equipped to model temporal dynamics or biochemical dependencies across multivariate time-series.

In contrast, STT-LLM correctly identified the anomaly and generated a structured explanation grounded in structural-temporal relationships (Fig. 14). It reasoned that the value of metabolite T (29.0) in Sample 4 was significantly higher than in the other samples (range 12-20). It further justified that normal metabolic progression could not explain this deviation. Instead, it correctly attributed the anomaly to potential doping. This capability highlights the value of STT-LLM's embedding-guided tokenization in aligning metabolite patterns over time and across variables, something missing in models lacking structured input alignment. Importantly, this structural-temporal reasoning also allows STT-LLM to avoid false positives in edge cases, such as when all samples from a subject appear uniformly elevated due to a legitimate medical condition. While other LLMs might misclassify such cases as doping, STT-LLM has the capacity to model intra-profile consistency, thereby reducing bias and increasing interpretability in high-stakes clinical scenarios.

**Phi**

1) The anomalous sample is Sample 9.

2) The reason is that Sample 9 shows a significantly higher concentration of metabolites Adiol, Bdiol, A, Etio, and T, which deviates from the expected pattern of steroid metabolism.

3) The anomaly is likely because the sample does not belong to the same athlete, as the metabolite concentrations are inconsistent with the rest of the profile.

6) The anomaly suggests the presence of a sample from a different source rather than a pattern of steroid use.

**Llama3**

1. The anomalous sample is the one where Adiol is much higher than the other steroid metabolites, which is sample 4.

2. The reason for the anomaly in sample 4 is likely due to the high levels of Etio and Adiol, indicating that the athlete may have been taking anabolic steroids or undergoing intense steroid use, causing an imbalance in the steroid metabolism pathway.

3. The anomaly is likely due to steroid doping because the high levels of Etio and Adiol are not typical of a natural steroid profile.

**DeepSeek**

Maybe the issue is that the diols are lower than expected? Or that T is low relative to A, which would be a problem because normally, T should be higher than A if there's metabolism. Or wait, A is the precursor to T, so A should be lower when T is high. If A is high and T is low, that would be unusual. So in samples 1 and 2, A is high, T is low. The same for sample 3: A is 820, T is 12, so A/T is about 68:1. Again, A is much higher than T.

**Mistral**

1) Sample 4

2) Anomaly in Sample 4: In sample 4 the concentration of Bdiol is 160 ug/ml which is not consistent with the expected concentration of Bdiol based on the steroid metabolism pathway. The concentration of Bdiol should be 1.6ug/ml (160/100) and hence the sample is anomalous.

**Falcon**

1) Sample 9: The metabolite A is 3000.0 and the metabolite Etio is 2100.0. The metabolite A should be higher than the metabolite Etio as it comes after the Bdiol in the pathway

2) The sample number of the anomalous sample is 9

3) I think this sample is due to steroid doping

**Qwen**

Q: 1)Sample 9 is the anomalous sample within the profile.

2) A: Sample 9 shows an unusually high value for both Adiol and Bdiol, indicating a possible metabolic anomaly or potential steroid doping.

3) The anomaly in sample 9 is most likely due to steroid doping, as the unusually high values for Adiol and Bdiol suggest an external source of these metabolites.

*Figure 13.* Contextual explanations from different models.

**STT-LLM**

## Step 1: Identify the anomalous sample

Sample 4 is the anomalous sample because it shows a significantly higher concentration of T (29.0) compared to the other samples, which are generally in the range of 12-20.

## Step 2: Reason for the anomaly based on steroid metabolism pathway

The anomaly could be due to the athlete being under steroid doping, as the concentration of T is significantly higher than the other samples. In a normal steroid profile, the concentration of T would decrease over time due to metabolism, but in this case, it remains high.

## Step 3: Reason for the anomaly

The anomaly is likely due to steroid doping, as the high concentration of T is not consistent with the normal metabolism of steroids in the body. The final answer is: 4

*Figure 14.* Contextual explanations from STT-LLM model.

