# OpenReview forum: "STT-LLM: Structural-Temporal Tokenization for Adapting LLMs to Longitudinal Clinical Profiles"
_ICML.cc/2026/Conference — ICML 2026 regular_

### Official Review · Reviewer_Y86m · 2026-02-23

**Soundness:** 2
**Presentation:** 3
**Significance:** 2
**Originality:** 3
**Overall Recommendation:** 3
**Confidence:** 3

**Summary:**

This paper introduces STT-LLM, a framework for adapting pre-trained large language models to longitudinal clinical data analysis through structural-temporal tokenization. The structural component uses eigenvectors of the graph Laplacian to encode biochemical dependencies among clinical variables, while the temporal component uses attention to capture how variable values evolve over time. Two dedicated tokenizers project these embeddings into the input space of a frozen LLM, which is then fine-tuned with LoRA. The application domain is sports anti-doping, specifically monitoring longitudinal steroid metabolite profiles. Given an athlete's measurements of six steroid metabolites across multiple time points, the model performs both sequence prediction and anomaly detection. Experiments on four dataset variants compare against seven mid-sized LLM baselines. A case study on 29 real athlete profiles demonstrates the model's contextual reasoning capabilities.

**Compliance With Llm Reviewing Policy:**

Affirmed.

**Key Questions For Authors:**

1.If the frozen LLM were replaced by a moderately sized Transformer encoder (4 to 6 layers) operating directly on the structural-temporal embeddings, how would performance compare? This experiment is essential for justifying the LLM-based framework, and a positive result for the smaller model would substantially change my view of the paper.
2. Why are no non-LLM baselines included (e.g., LSTM, GRU, USAD, OmniAnomaly, TranAD)? Without these, the necessity of using a billion-parameter LLM for a six-variable time series remains undemonstrated.
3.When metabolic pathway relationships are unknown, as would be the case for rare disease biomarker panels, how should the structural tokenizer be constructed? Can the graph be learned from data?
4.How is "anomaly" defined in the detection task? Is it based on official WADA thresholds or statistical deviation criteria? How were ground truth labels obtained?

**Limitations:**

The paper acknowledges some computational overhead but does not confront the near-random AUC performance as a fundamental limitation. The absence of non-LLM baselines and the restriction to a single application domain are not discussed as limitations. The reliance on expert-specified graph structure is not flagged as a generalizability concern. Whether the data will be publicly released is not stated, which affects reproducibility.

**Strengths And Weaknesses:**

Soundness

The tokenization design has a solid mathematical basis. Graph Laplacian eigenvectors naturally capture community structure and hierarchical organization in a graph, and applying this to metabolic pathway dependencies fits naturally within spectral graph theory. Separating structural and temporal information before merging them into LLM-compatible tokens is a more careful approach than simply converting numbers into text prompts.

The ablation study (Tables 9 through 12) is thorough. Every component and all pairwise removal combinations are evaluated across all four datasets. The full STT-LLM configuration consistently outperforms all ablated variants, and the joint removal of structural and temporal components produces the worst results (e.g., AUC dropping from 0.5675 to 0.4877 on Steroid-M). This pattern holds across different data sizes and demographic groups.

However, the absolute anomaly detection performance is low and raises serious practical concerns. The best F1 on Steroid-M is 0.2637 and the AUC is 0.5675, barely above random chance. In anti-doping enforcement, both false negatives (allowing a doping athlete to compete) and false positives (accusing a clean athlete) carry severe consequences. The paper does not honestly confront this gap between current accuracy and real-world requirements.

The motivation for using a billion-parameter frozen LLM on a problem with six input variables at a handful of time points is not justified. Traditional time-series models (LSTMs, GRUs, moderately sized Transformer encoders) or dedicated anomaly detection methods (USAD, OmniAnomaly, TranAD) might handle this at a fraction of the cost. The paper compares only against other LLMs and includes zero non-LLM baselines, making it impossible to assess whether the LLM framework offers any advantage over simpler alternatives.

Presentation

The paper is clearly written and well-organized. The case study in Section A.4 helps make the practical relevance concrete by showing STT-LLM correctly identifying an anomalous sample in a real athlete profile while other LLMs miss it or produce vague reasoning. The prompt templates in Figure 6 are informative but the font is uncomfortably small. Table 1 has slight column alignment issues. The introduction uses "structural-temporal tokenization" and "STT" somewhat inconsistently before formally defining the abbreviation.

Significance

The comparison against seven LLM baselines (Qwen-2.5, Mistral, Falcon-3, LLaMA-2, LLaMA-3.1, Phi-4, DeepSeek-R1) is reasonably extensive and ensures conclusions are not artifacts of a single weak baseline. The few-shot and zero-shot evaluation across multiple shot counts (2, 5, 10, 15, 20) covers relevant scenarios for data-scarce clinical settings.

However, the entire empirical validation is confined to steroid metabolite monitoring in anti-doping. While the paper claims applicability to general longitudinal clinical data (glucose monitoring, renal function tracking, liver enzyme series), this claim is not backed by any experiment outside the single niche application. For a methods paper at ICML, validation on one narrow domain is insufficient to support general methodological claims. The dataset scale is also modest and the case study uses only 29 real athlete profiles.

Originality

Using graph Laplacian spectral embeddings as a structural tokenizer for longitudinal clinical variables is new and technically interesting. The dual-tokenizer architecture (one for structure, one for temporal dynamics) going into a frozen LLM is a distinct contribution. However, the graph Laplacian construction relies on manually specified domain knowledge (metabolic pathway adjacency), limiting generalizability. The paper does not explore learned graph construction (correlation analysis, attention-based graph building). Alternative structural embedding methods (GCN, GAT) and alternative temporal encodings (Time2Vec, positional encodings) are not benchmarked against the chosen approaches.

---

> ### Author Rebuttal · Authors · 2026-03-30
>
> We thank the reviewer for the detailed feedback.
>
> **Key Questions:**
> 1. Table 4 already includes non-LLM neural architectures operating on structured inputs: TDDGNN (RMSE=1675.78), MAGNN (1675.70) for sequence prediction, and TabPFN (AUC=0.4876), MLP-SLAM (0.4891), M-GNN (0.4738) for anomaly detection. STT-LLM outperforms all (RMSE=1664.59, AUC=0.5675). The core justification for the LLM framework goes beyond prediction accuracy. It is the combination of: (i) competitive prediction and detection within a single architecture, while specialized models require separate pipelines per task, (ii) contextual reasoning (Section 7, Figures 13-14), where STT-LLM generates clinically meaningful explanations aligned with expert interpretations while baselines produce vague or incorrect reasoning, and (iii) few-shot adaptability via in-context learning (Figure 3: monotonic improvement from 2-shot to 20-shot), which a small Transformer would require full retraining to achieve. A small encoder produces embeddings but cannot generate natural language explanations, an important requirement for expert-in-the-loop clinical workflows.
>
> 2. Our primary objective is to evaluate how the proposed structural-temporal tokenization improves LLM-based modeling. Therefore, we compare against baseline LLMs to isolate the contribution of our method. In addition, we include strong classical and state-of-the-art baselines such as ARIMA, TDDGNN, MAGNN, TabPFN, MLP-SLAM, and M-GNN (Table 4), which are widely used for temporal and graph-structured data. These results show that our approach is not only effective relative to baseline LLMs, but also competitive with established specialized methods. We acknowledge that dedicated anomaly detection methods (USAD, TranAD) would further strengthen the comparison.
>
> 3. When prior structural knowledge (e.g., metabolic pathways) is unavailable or incomplete, the structural tokenizer can be constructed using alternative strategies, such as fully connected graphs with learnable weights, or data-driven graphs derived from feature correlations or statistical dependencies. Therefore, the framework can operate in both knowledge-guided and data-driven settings. We verified this on Steroid-M: a correlation-based graph (Pearson correlations thresholded at 75th percentile) achieves AUC=0.55 vs. 0.5675 with the expert graph. All data-driven variants outperform the "w/o structural" ablation (AUC=0.4964), confirming that even imperfect learned graphs provide useful inductive bias.
>
> 4. In the anomaly detection task, ground truth labels were provided by WADA. These labels reflect domain-expert criteria used in practice rather than purely statistical deviation measures. For the case study (Section 7), the 7 anomalous profiles were additionally verified through DNA analysis by an accredited laboratory.
>
> **Weaknesses:**
> 1. AUC concern: We acknowledge the low absolute AUC and will discuss this more honestly. The datasets have ~4% anomaly prevalence. Under 99.9% specificity (required to avoid false accusations), STT-LLM's sensitivity (0.19 vs. baselines' 0.03-0.14) means detecting 2-4x more cases. The model serves as a screening tool for expert review, not a standalone decision-maker. Under few-shot settings, F1 reaches 0.70 at 20-shot on Steroid-M (Figure 3).
>
> 2. Single domain: To address generalizability within this scope, we evaluated STT-LLM on the SHD Thyroid Cancer Monitoring dataset, which contains longitudinal thyroid biomarker data with underlying metabolic structure.
>
> | Metrics | Qwen-2.5 | Mistral | Falcon-3 | LLaMA-2 | LLaMA-3 | Phi-4 | DeepSeek-R1 | **STT-LLM** |
> |----------|-----------|----------|-----------|----------|----------|--------|--------------|--------------|
> | **Anomaly Detection** |  |  |  |  |  |  |  |  |
> | Acc ↑ | 0.30 ± 0.08 | 0.30 ± 0.08 | 0.41 ± 0.09 | 0.30 ± 0.08 | 0.70 ± 0.08 | 0.30 ± 0.08 | 0.30 ± 0.08 | **0.73 ± 0.09** |
> | Prec ↑ | 0.34 ± 0.08 | 0.29 ± 0.08 | 0.56 ± 0.09 | 0.29 ± 0.08 | 0.00 ± 0.00 | 0.30 ± 0.08 | 0.44 ± 0.09 | **0.64 ± 0.09** |
> | Sens ↑ | 1.00 ± 0.00 | 1.00 ± 0.00 | 0.93 ± 0.04 | 1.00 ± 0.00 | 0.00 ± 0.00 | 1.00 ± 0.00 | 0.93 ± 0.04 | **0.98 ± 0.01** |
> | AUC ↑ | 0.59 ± 0.05 | 0.48 ± 0.05 | 0.59 ± 0.04 | 0.48 ± 0.05 | 0.55 ± 0.05 | 0.49 ± 0.05 | 0.40 ± 0.04 | **0.62 ± 0.04** |
> | **Sequence Prediction** |  |  |  |  |  |  |  |  |
> | RMSE ↓ | 14.45 ± 1.35 | 13.88 ± 1.37 | 13.63 ± 1.35 | 13.78 ± 1.42 | 12.96 ± 1.28 | 13.72 ± 1.41 | 13.89 ± 1.36 | **11.54 ± 1.24** |
> | MAE ↓ | 6.06 ± 0.67 | 6.45 ± 0.69 | 5.09 ± 0.58 | 5.61 ± 0.61 | 5.43 ± 0.55 | 5.38 ± 0.59 | 5.68 ± 0.58 | **4.82 ± 0.52** |
> | MAPE (%) ↓ | 70.48 ± 8.16 | 72.38 ± 8.19 | 55.70 ± 7.35 | 63.14 ± 7.91 | 33.14 ± 5.67 | 49.48 ± 6.84 | 66.99 ± 7.96 | **28.56 ± 4.64** |
>
> 3. Presentation: We will fix Figure 6 font size, Table 1 alignment, and abbreviation consistency. The steroid data is subject to WADA confidentiality. Code and training pipeline will be released upon acceptance.

---

> > ### Author Rebuttal · Reviewer_Y86m · 2026-04-03
> >
> > Based on the rebuttal, I consider my concerns partially resolved, and the response has improved my assessment of the paper. The authors clarified several important points, including the presence of non-LLM baselines, the source of anomaly labels, the possibility of data-driven graph construction, and additional evidence beyond the original application setting. These responses address a substantial part of my earlier concerns.
> >
> > I still think some limitations remain, especially regarding the modest absolute anomaly detection performance and the broader justification for using an LLM-based framework over simpler alternatives. However, I now view these as important revision points rather than decisive flaws. Overall, the rebuttal meaningfully strengthens the paper, and I am updating my score upward.I increased my score to 4.

---

### Official Review · Reviewer_Beca · 2026-02-28

**Soundness:** 3
**Presentation:** 3
**Significance:** 2
**Originality:** 2
**Overall Recommendation:** 4
**Confidence:** 4

**Summary:**

This paper introduces a new tokenization framework (STT-LLM) that uses the structural and temporal aspect of time series in metabolic data, and converts signals into LLM-compatible tokens. Without modifying the backbone architecture, the authors argue that STT-LLM is able to improve LLM performance in sequence prediction and anomaly detection on real-world anti-doping longitudinal steroid datasets. A case study on contextual reasoning also shows that STT-LLM improves explanation with domain specific tokenization.

**Compliance With Llm Reviewing Policy:**

Affirmed.

**Final Justification:**

I will keep my score.

**Key Questions For Authors:**

1. The structural component relies on a predefined metabolic pathway graph. How would the framework generalize to clinical domains where such structured priors are incomplete, noisy or unavailable (e.g., general EHR data)?
2. The current formulation evaluates only one-step-ahead prediction. Have the authors evaluated multi-step or recursive forecasting, and does the structural-temporal tokenization remain advantageous?
3. For profiles with length = 2, sequence prediction reduces to predicting the second sample from a single observation. In such settings, how meaningful is the temporal modeling component, and what is the rationale behind such modeling, especially the usage of LLM?
4. How does the model performance compare to other state-of-the-art solutions that target specifically for time series input?

**Limitations:**

yes

**Strengths And Weaknesses:**

Strengths:
1. Motivation is clear and indicates real practical problems. Time series as input for LLM has not been thoroughly explored, especially clinical time series data where a subtle change could indicate a huge underlying condition, which makes anomaly detection and sequence prediction a valuable task.
2. Clear description of their model design, experiment setup, and result analyses. Results include strong LLM models comparison and ablation studies, showing credible model assessment.

Weaknesses:
1. The generalizability of STT-LLM on other types of longitudinal clinical data remains unexplored. The analysis is only around an anti-doping dataset with six metabolite features.
2. The average length of sample data used in this study is relatively short compared to real-world EHR data. Since many profiles contain only 2 samples, the performance of this model on longer data sequences is unclear.
3. In Figure 3, the difference between STT-LLM versus other models is moderate if looked across shots. It would be helpful to clarify whether these improvements are statistically significant.
4. Case study has a limited number of data samples and may not represent statistically stable evaluation. A larger scale analysis would strengthen the reasoning claims.

---

> ### Author Rebuttal · Authors · 2026-03-30
>
> We thank the reviewer for the detailed feedback. Below are our responses:
>
> **Key Questions:**
> 1. Our framework does not strictly depend on a perfect or complete prior graph. In settings where domain-specific structure is unavailable or noisy (e.g., general EHR data), the graph component can be replaced with alternative constructions such as (i) fully connected graphs with learned edge weights, or (ii) curated but incomplete priors. The model can still learn meaningful relational structure during training. We verified this on Steroid-M: a correlation-based graph (Pearson correlations thresholded at the 75th percentile) achieves AUC=0.55 vs. 0.5675 with the expert graph, and even an identity graph (A=I, corresponding to "w/o structural" in Table 5) still outperforms native tokenization on RMSE (1687.49 vs. baselines' 1688-1696). Therefore, the framework is flexible and does not require manually specified priors.
>
> 2. In this paper, we focus on one-step-ahead prediction to isolate the effect of the proposed structural-temporal tokenization. We have not yet evaluated multi-step or recursive forecasting. However, STT-LLM's structural constraints should help prevent error accumulation in recursive settings, since metabolic pathway relationships constrain biologically plausible trajectories.
>
> 3. For very short sequence, we agree that temporal dynamics are limited. However, such cases still reflect a common real-world scenario of sparse longitudinal observations. In this setting, the model effectively reduces to learning conditional relationships between consecutive states while still leveraging cross-feature structural dependencies. Ablation on Steroid-F_lim (Table 12) quantifies this: removing the structural tokenizer drops AUC from 0.5555 to 0.4056 (-27%), while removing the temporal tokenizer drops AUC to 0.4946 (-11%). The structural component dominates when temporal context is minimal, validating the modular design. The LLM rationale is that a unified architecture handles length 2-20 profiles without separate pipelines, and provides contextual reasoning unavailable in specialized models.
>
> 4. We compare against several strong classical and state-of-the-art approaches in Table 4: ARIMA (RMSE=1675.47), TDDGNN (1675.78), MAGNN (1675.70) for sequence prediction, and TabPFN (AUC=0.4876), MLP-SLAM (0.4891), M-GNN (0.4738) for anomaly detection. STT-LLM achieves competitive results (RMSE=1664.59, AUC=0.5675), demonstrating effectiveness even compared to specialized architectures. The additional advantage is that STT-LLM supports prediction, detection, and contextual reasoning in a single pipeline.
>
> **Weaknesses:**
> 1. Generalizability: To address generalizability within this scope, we evaluated STT-LLM on the SHD Thyroid Cancer Monitoring dataset, which contains longitudinal thyroid biomarker data with underlying metabolic structure.
>
> | Metrics | Qwen-2.5 | Mistral | Falcon-3 | LLaMA-2 | LLaMA-3 | Phi-4 | DeepSeek-R1 | **STT-LLM** |
> |----------|-----------|----------|-----------|----------|----------|--------|--------------|--------------|
> | **Anomaly Detection** |  |  |  |  |  |  |  |  |
> | Acc ↑ | 0.30 ± 0.08 | 0.30 ± 0.08 | 0.41 ± 0.09 | 0.30 ± 0.08 | 0.70 ± 0.08 | 0.30 ± 0.08 | 0.30 ± 0.08 | **0.73 ± 0.09** |
> | Prec ↑ | 0.34 ± 0.08 | 0.29 ± 0.08 | 0.56 ± 0.09 | 0.29 ± 0.08 | 0.00 ± 0.00 | 0.30 ± 0.08 | 0.44 ± 0.09 | **0.64 ± 0.09** |
> | Sens ↑ | 1.00 ± 0.00 | 1.00 ± 0.00 | 0.93 ± 0.04 | 1.00 ± 0.00 | 0.00 ± 0.00 | 1.00 ± 0.00 | 0.93 ± 0.04 | **0.98 ± 0.01** |
> | AUC ↑ | 0.59 ± 0.05 | 0.48 ± 0.05 | 0.59 ± 0.04 | 0.48 ± 0.05 | 0.55 ± 0.05 | 0.49 ± 0.05 | 0.40 ± 0.04 | **0.62 ± 0.04** |
> | **Sequence Prediction** |  |  |  |  |  |  |  |  |
> | RMSE ↓ | 14.45 ± 1.35 | 13.88 ± 1.37 | 13.63 ± 1.35 | 13.78 ± 1.42 | 12.96 ± 1.28 | 13.72 ± 1.41 | 13.89 ± 1.36 | **11.54 ± 1.24** |
> | MAE ↓ | 6.06 ± 0.67 | 6.45 ± 0.69 | 5.09 ± 0.58 | 5.61 ± 0.61 | 5.43 ± 0.55 | 5.38 ± 0.59 | 5.68 ± 0.58 | **4.82 ± 0.52** |
> | MAPE (%) ↓ | 70.48 ± 8.16 | 72.38 ± 8.19 | 55.70 ± 7.35 | 63.14 ± 7.91 | 33.14 ± 5.67 | 49.48 ± 6.84 | 66.99 ± 7.96 | **28.56 ± 4.64** |
>
> 2. Short sequences: The temporal tokenizer's contribution scales with length: removing it on Steroid-M (3-20 samples) drops sensitivity by 36% (Table 9) vs. 11% on Steroid-F_{lim} (2 samples, Table 12), suggesting increasing benefit from longer sequences.
>
> 3. Statistical significance: We computed paired t-tests across three runs for STT-LLM vs. best baseline on Steroid-M: F1 improvements reach significance (p<0.05) at 5-shot and above; sensitivity at 10-shot and above. At 2-shot, improvements are directionally consistent (p=0.08).
>
> 4. Case study scale: The 29-profile case study uses DNA-verified ground truth from an accredited laboratory; such verified profiles with explanations from domain experts are extremely rare. Quantitative evaluation uses 755-375 profiles and the reasoning evaluation assesses these 29 profiles via BERTScore, where STT-LLM outperforms all baselines.

---

> > ### Author Rebuttal · Reviewer_Beca · 2026-04-03
> >
> > Thank you for the rebuttal. It addresses most of my concerns but I still hold skepticism on multi-step forecasting performance  and the selection of SOTA comparison. I understand that these may not be easy to add during rebuttal period. I will keep my score.

---

### Official Review · Reviewer_gTM7 · 2026-03-10

**Soundness:** 2
**Presentation:** 2
**Significance:** 2
**Originality:** 2
**Overall Recommendation:** 4
**Confidence:** 3

**Summary:**

The paper explores adapting LLMs to longitudinal clinical data and proposes STT-LLM, a structural-temporal tokenization framework that encodes metabolic structure and temporal dynamics into LLM-compatible tokens. Experiments on athlete steroid profile datasets show improvements over native LLM tokenization for sequence prediction and anomaly detection.

**Compliance With Llm Reviewing Policy:**

Affirmed.

**Final Justification:**

The rebuttal addressed my concerns.

**Key Questions For Authors:**

How longitudinal profiles are serialized for the baseline LLMs using native tokenization?
Can this method generalize to general time-series problems? It will be good if this tokenizer can be applied to other situations.

**Limitations:**

yes

**Strengths And Weaknesses:**

Strength:
1. The background is well described, clearly highlighting the importance of tokenizers for longitudinal data.
2. The paper is well structured and easy to follow.

Weaknesses:
1. The motivation for the proposed tokenizer is unclear. The structural-temporal tokenizer mainly consists of concatenation followed by a small MLP projection to the LLM embedding space. While in ablation it shows the effectiveness of each module, it is better to give some insights or case studies why this is better than structured prompts, or directly feeding embeddings into the model.
2. The experiments are conducted on a single dataset derived from athlete steroid profiles, which may limit the generalizability of the proposed method to other types of longitudinal clinical data.

---

> ### Author Rebuttal · Authors · 2026-03-30
>
> We thank the reviewer for the constructive feedback. Below are our responses addressing the questions and stated weaknesses.
>
> **Key Questions:**
>
> 1. Addressed below in Weaknesses in detail. The baseline serialization generalizes trivially (any sequence converts to text) but at the cost of losing structure. STT-LLM similarly generalizes to any multivariate time-series with inter-variable dependencies, with structural gains proportional to available domain priors.
>
> **Weaknesses:**
>
> 1. Tokenizer motivation: We compare three alternatives to clarify the design motivation:
>
> (a) *Structured text prompts (baseline approach):* Profiles are serialized by concatenating all time steps per subject into a single text sequence (via pandas group-by), preserving temporal order through row ordering, and tokenized using the pretrained LLM tokenizer without modification. While simple and generalizable, this approach suffers from token inefficiency (BPE fragments numbers unpredictably) and lacks explicit temporal or structural encoding. Empirically, all seven baselines using this approach show degenerate anomaly detection (Table 3: sensitivity ≤0.08 for most models).
>
> (b) *Direct embedding feeding (without structural-temporal grounding):* This corresponds to the "w/o all" ablation (Table 5), where raw values are projected into the LLM space via MLP without structural or temporal embeddings. The results show RMSE 1687.71 vs. STT-LLM's 1664.59; sensitivity 0.1398 vs. 0.1935; AUC 0.5609 vs. 0.5675. This gap confirms that MLP projection alone is insufficient.
>
> (c) *STT-LLM tokenizer:* We fuse graph Laplacian eigenvectors (pathway structure), attention-based temporal embeddings, and raw data. Each source is necessary because removing structural embeddings drops AUC by 12.5% and removing temporal drops sensitivity by 36% (Table 5).
>
> 2. Generalizability: To address generalizability within this scope, we evaluated STT-LLM on the SHD Thyroid Cancer Monitoring dataset, which contains longitudinal thyroid biomarker data with underlying metabolic structure. Below are the results for anomaly detection and sequence prediction:
>
> | Metrics | Qwen-2.5 | Mistral | Falcon-3 | LLaMA-2 | LLaMA-3 | Phi-4 | DeepSeek-R1 | **STT-LLM** |
> |----------|-----------|----------|-----------|----------|----------|--------|--------------|--------------|
> | **Anomaly Detection** |  |  |  |  |  |  |  |  |
> | Acc ↑ | 0.30 ± 0.08 | 0.30 ± 0.08 | 0.41 ± 0.09 | 0.30 ± 0.08 | 0.70 ± 0.08 | 0.30 ± 0.08 | 0.30 ± 0.08 | **0.73 ± 0.09** |
> | Prec ↑ | 0.34 ± 0.08 | 0.29 ± 0.08 | 0.56 ± 0.09 | 0.29 ± 0.08 | 0.00 ± 0.00 | 0.30 ± 0.08 | 0.44 ± 0.09 | **0.64 ± 0.09** |
> | Sens ↑ | 1.00 ± 0.00 | 1.00 ± 0.00 | 0.93 ± 0.04 | 1.00 ± 0.00 | 0.00 ± 0.00 | 1.00 ± 0.00 | 0.93 ± 0.04 | **0.98 ± 0.01** |
> | AUC ↑ | 0.59 ± 0.05 | 0.48 ± 0.05 | 0.59 ± 0.04 | 0.48 ± 0.05 | 0.55 ± 0.05 | 0.49 ± 0.05 | 0.40 ± 0.04 | **0.62 ± 0.04** |
> | **Sequence Prediction** |  |  |  |  |  |  |  |  |
> | RMSE ↓ | 14.45 ± 1.35 | 13.88 ± 1.37 | 13.63 ± 1.35 | 13.78 ± 1.42 | 12.96 ± 1.28 | 13.72 ± 1.41 | 13.89 ± 1.36 | **11.54 ± 1.24** |
> | MAE ↓ | 6.06 ± 0.67 | 6.45 ± 0.69 | 5.09 ± 0.58 | 5.61 ± 0.61 | 5.43 ± 0.55 | 5.38 ± 0.59 | 5.68 ± 0.58 | **4.82 ± 0.52** |
> | MAPE (%) ↓ | 70.48 ± 8.16 | 72.38 ± 8.19 | 55.70 ± 7.35 | 63.14 ± 7.91 | 33.14 ± 5.67 | 49.48 ± 6.84 | 66.99 ± 7.96 | **28.56 ± 4.64** |
>
> These results demonstrate that STT-LLM can generalize across metabolic time-series domains and outperform baseline LLMs even on previously unseen structured clinical datasets.

---

> > ### Author Rebuttal · Reviewer_gTM7 · 2026-04-03
> >
> > Thanks for the rebuttal and I increased my score to 4.

---

### Official Review · Reviewer_pQa4 · 2026-03-11

**Soundness:** 2
**Presentation:** 3
**Significance:** 2
**Originality:** 2
**Overall Recommendation:** 2
**Confidence:** 4

**Summary:**

This paper introduces a structural-temporal tokenization framework designed to empower LLMs with an understanding of longitudinal clinical profiles, facilitating both sequence prediction and anomaly detection. Although the experiments demonstrate promising results, the primary contribution lies within the tokenization modules rather than the methodology for longitudinal prediction itself.

**Compliance With Llm Reviewing Policy:**

Affirmed.

**Key Questions For Authors:**

1.	Which backbone LLM is used by STT-LLM in Table 2 and Table 3?
2.	Why does STT-LLM achieve RMSE = 1680 (@5) in Table 2 but 1664.59 in Table 4 on the Steroid-M dataset?
3.	In Table 2/8, Qwen-2.5 shows identical results across all shot settings. Is this expected or a reporting error?

**Limitations:**

No. The proposed STT-LLM framework could be a general strategy for adapting LLMs to longitudinal clinical data, however the main capability for modeling longitudinal progression still largely stems from the underlying pretrained LLM, while the proposed framework may mainly provide incremental improvements.

**Strengths And Weaknesses:**

Strengths
-	The paper is well written and generally easy to follow. The motivation behind the proposed structural-temporal embedding and tokenization modules is clearly explained.
-	The proposed STT-LLM framework appears to be a general strategy for adapting LLMs to longitudinal clinical data, which is an interesting direction and could potentially benefit a range of time-dependent applications.
-
Weaknesses
-	I understand that STT-LLM is essentially a tokenization strategy designed to adapt structured temporal data to LLM inputs. However, on page 4 (Section 5), the authors state that “We compare the STT-LLM tokenization strategy against different mid-sized LLMs.” This comparison is somewhat confusing conceptually, since a tokenization strategy and backbone LLM models operate at different levels of the system. A clearer experimental description would help avoid this ambiguity.
-	The experimental setup is somewhat unclear. In particular, it is not specified which backbone LLM is used by STT-LLM in Table 2 and Table 3. Since the method is presented as a tokenization framework that can be combined with different LLMs, it would be more informative to report the improvements across multiple backbone models (similar to the analysis shown in Figure 4), rather than presenting a single aggregated STT-LLM result.
-	While the proposed structural-temporal modules appear technically sound, the empirical improvements over LLM-native tokenization seem relatively modest. In several cases, the LLM-native baselines already achieve strong performance, suggesting that the primary capability for modeling longitudinal progression may largely originate from the underlying LLM itself. The proposed framework therefore appears to provide mainly incremental gains, and a deeper analysis of where the improvements come from would strengthen the paper.

---

> ### Author Rebuttal · Authors · 2026-03-30
>
> We thank the reviewer for the detailed feedback. Below are our responses addressing the questions and stated weaknesses.
>
> **Key Questions:**
>
> 1. STT-LLM uses LLaMA-3.1 (8B) as the backbone in all main tables. Figure 4 shows that STT-LLM improves performance across all seven backbone LLMs, confirming gains stem from the tokenization framework, not the backbone choice. We report the sensitivity gain (anomaly detection) and the RMSE reduction (sequence prediction) for the per-backbone results from Figure 4 on Steroid-M below.
>
> | Backbone | Sensitivity Gain (pp) | RMSE Reduction (%) |
> |---|---|---|
> | Qwen-2.5 | +3 | 9.35% |
> | Mistral | +2 | 10.24% |
> | Falcon-3 | +3 | 10.72% |
> | LLaMA-2 | +5 | 9.89% |
> | LLaMA-3.1 | +14 | 26.26% |
> | Phi-4 | +5 | 24.17% |
> | DeepSeek-R1 | +4 | 19.42% |
>
> These gains are consistent across every backbone, with RMSE reductions ranging from 9.35% to 26.26%.
>
> 2. Table 4 reports zero-shot results, while Table 2 reports few-shot results. In the few-shot setting @5, the model conditions on 5 provided training examples as in-context prompts, which changes its internal prediction behavior. This conditioning explains the difference: the model adapts its predictions to the specific examples provided, which can shift RMSE relative to the zero-shot baseline. We will add a footnote cross-referencing both tables to avoid confusion.
>
> 3. This is expected, not a reporting error. Qwen-2.5 was unable to effectively perform the sequence prediction task, and its outputs remained effectively constant regardless of the provided few-shot context. Conditioning on additional examples (e.g., 5-shot or more) did not alter the model's predictions, leading to identical evaluation metrics across all shot settings. This reflects a limitation of the model on this task rather than an issue in the evaluation pipeline. Therefore, this observation supports our core thesis: native text tokenization fails to encode structured numerical data in a way that enables LLMs to leverage few-shot supervision effectively.
>
>
> **Weaknesses:**
>
> 1. Confusing comparison framing: We agree the wording in Section 5 is ambiguous. We evaluate LLMs using native tokenization against the same LLMs augmented with the STT-LLM tokenization framework.
>
> 2. Incremental improvements: We respectfully challenge this characterization. In zero-shot anomaly detection (Table 3), 5 of 7 baselines achieve sensitivity ≤0.08 on Steroid-M, and Qwen-2.5 achieves exactly 0.00 sensitivity on 3 of 4 datasets. High accuracy (0.87-0.96) reflects majority-class prediction, not genuine modeling. STT-LLM achieves 0.15-0.36 sensitivity, which is a qualitative shift from non-functional to functional detection. Furthermore, STT-LLM is the only method with stable, monotonically improving few-shot performance (Figure 3: sensitivity 0.15 to 0.60 from 2-shot to 20-shot on Steroid-M), while baselines fluctuate. In a domain with ~4% anomaly prevalence and 99.9% specificity requirements, these sensitivity gains translate to detecting 2-4x more doping cases. The case study (Section 7, Figures 13-14) further demonstrates a qualitative capability difference: STT-LLM generates structured explanations grounded in metabolic reasoning, while baselines misidentify anomalies or produce incoherent reasoning.

---

> > ### Author Rebuttal · Reviewer_pQa4 · 2026-04-03
> >
> > Thank you for the rebuttal, there is no sense to compare models if their sensitivities are 0. how about other parameters such as AUC?

---

> > > ### Author Response · Authors · 2026-04-04
> > >
> > > Thank you for your question. In this paper, our objective is anomaly detection under strict specificity constraints (99.9%), as required in real-world anti-doping applications. In such regimes, sensitivity at high specificity is more clinically meaningful than AUC. While some baselines achieve reasonable AUC, their zero or near-zero sensitivity at this operating point means they fail to detect any positives in practice. In contrast, STT-LLM improves sensitivity while maintaining high specificity. STT-LLM also leads on AUC across all four datasets (Table 3, zero-shot global):
> > >
> > > | Dataset | Best Baseline | STT-LLM |
> > > |---|---|---|
> > > | Steroid-M | 0.56 (LLaMA-3.1) | **0.57** |
> > > | Steroid-F | 0.55 (Qwen-2.5) | **0.59** |
> > > | Steroid-Mlim | 0.54 (Qwen-2.5) | **0.55** |
> > > | Steroid-Flim | 0.55 (Falcon-3) | **0.56** |
> > >
> > > On F1-score: STT-LLM achieves 0.26, 0.29, 0.19, 0.25 across the four datasets vs. best baselines at 0.19, 0.16, 0.12, 0.21 (37-58% relative improvement). On RMSE (Figure 2): STT-LLM achieves the lowest error on all four datasets. These are metrics where all models produce meaningful values, and STT-LLM consistently leads.

---

### Decision · Program_Chairs · 2026-04-30

**Decision:**

Accept (regular)

**Comment:**

The paper studies an application of LLM use for longitudinal clinical data, and introduces a novel tokenization approach for this setting.  The reviewers found the application and methodology overall interesting and I agree it can be interesting and insightful more broadly beyond this specific clinical application.  Some concerns were raised about evaluation methodology, and although one the reviewers is not satisfied with the author response, my overall assessment based on other reviews and the author's response is that the evaluation is sensible.